# Reinforcing Spatial Reasoning in Vision-Language Models with Interwoven Thinking and Visual Drawing

**Junfei Wu**[123]*, **Jian Guan**[3]*, **Kaituo Feng**[4], **Qiang Liu**[12], **Shu Wu**[12]†, **Liang Wang**[12], **Wei Wu**[3]†, **Tieniu Tan**[125]

[1]NLPR, MAIS, Institute of Automation, Chinese Academy of Sciences.
[2]University of Chinese Academy of Sciences. [3]Ant Group.
[4]CUHK MMLab. [5]Nanjing University.
junfei.wu@cripac.ia.ac.cn, shu.wu@nlpr.ia.ac.cn,
{jianguanthu, wuwei19850318}@gmail.com

 Code: https://github.com/AntResearchNLP/ViLaSR
 Model: https://huggingface.co/AntResearchNLP/ViLaSR

## Abstract

As textual reasoning with large language models (LLMs) has advanced significantly, there has been growing interest in enhancing the multimodal reasoning capabilities of large vision-language models (LVLMs). However, existing methods primarily approach multimodal reasoning in a straightforward, text-centric manner, where both reasoning and answer derivation are conducted purely through text, with the only difference being the presence of multimodal input. As a result, these methods often encounter fundamental limitations in spatial reasoning tasks that demand precise geometric understanding and continuous spatial tracking—capabilities that humans achieve through mental visualization and manipulation. To address the limitations, we propose drawing to reason in space, a novel paradigm that enables LVLMs to reason through elementary drawing operations in the visual space. By equipping models with basic drawing operations, including annotating bounding boxes and drawing auxiliary lines, we empower them to express and analyze spatial relationships through direct visual manipulation, meanwhile avoiding the performance ceiling imposed by specialized perception tools in previous tool-integrated reasoning approaches. To cultivate this capability, we develop a three-stage training framework: cold-start training with synthetic data to establish basic drawing abilities, reflective rejection sampling to enhance self-reflection behaviors, and reinforcement learning to directly optimize for target rewards. Extensive experiments demonstrate that our model, named **ViLaSR**, consistently outperforms existing methods across diverse spatial reasoning benchmarks, involving maze navigation, static spatial reasoning, video-based reasoning, and multi-view-based reasoning tasks, with an average improvement of 18.4%. Ablation studies reveal the critical role of each training stage, where reflective rejection sampling strengthens the model's self-correction capabilities, and reinforcement learning effectively unlocks its reasoning potential.

## 1 Introduction

Large language models (LLMs) have exhibited remarkable reasoning capabilities in complex tasks such as mathematical problem-solving [66, 35] and code generation [6], particularly through the

---

*Equal Contribution. †Corresponding Author.

39th Conference on Neural Information Processing Systems (NeurIPS 2025).

"slow thinking" paradigm exemplified by OpenAI o1 [26] and DeepSeek R1 [10], which enables extended reasoning with in-depth self-reflection. Encouraged by the success, a growing body of research is now aiming to adapt similar techniques to large vision-language models (LVLMs) to enhance their capabilities in image and video reasoning [72, 11, 50, 13].

While current LVLMs excel at basic visual perception tasks such as object detection [12] and narrative understanding [40, 9], they often struggle with spatial reasoning [64, 70], which requires understanding spatial relationships among objects and tracking their dynamic evolution [60]—capabilities that are crucial for real-world applications such as robotics [49] and augmented reality [17]. Indeed, even when such deep visual understanding is required, LVLMs still rely solely on text-based reasoning [72], assuming that visual information can be perfectly translated into textual semantic space [23]. Unfortunately, such translation is inherently challenging [33]: spatial details are inevitably lost when converting visual information to text, and describing dynamic changes of object positions becomes prohibitively complex in textual space (see examples of GPT-4o in Figure 1). Taking inspiration from human cognition, where spatial reasoning relies on mental visualization and dynamic manipulation [3], we advocate a vision-centric reasoning paradigm where models actively edit and re-encode visual information at each reasoning step, dynamically supplementing spatial details and relationships. This "thinking with images" approach, while validated by OpenAI o3 [47], remains underexplored in open-source research.

In parallel with our work, recent studies have begun to integrate visual tools to enable vision-centric reasoning [8, 51, 21, 31]. However, these approaches exhibit limitations along two key dimensions. First, the reasoning capabilities of LVLMs are constrained by black-box perception tools (e.g., grounding and OCR systems), resulting in not only fixed perception capabilities but also fragmented reasoning composed of disconnected tool invocations that undermine coherence and holistic planning. Second, these methods heavily draw upon reasoning data curated based on human priors, which often exhibit oversimplified logic compared to the complexity of spatial reasoning tasks, with problem decomposition and tool invocation interleaved in a simplistic and linear fashion [8, 51]. Such adherence to prescribed reasoning patterns prevents LVLMs from developing the capacity for critical reflection on tool outputs—a capability that has proven crucial for advanced reasoning [45]. These fundamental limitations call for a more flexible and intrinsic approach to vision-centric reasoning.

In light of these challenges, we propose "drawing to reason in space," a novel, versatile reasoning paradigm that empowers LVLMs to reason through elementary drawing operations, as exemplified in Figure 1. Through simple yet powerful operations, including bounding boxes for object localization and auxiliary lines for relationship analysis, this paradigm enables direct visual interaction for spatial problem-solving, mirroring human behavior while avoiding the pitfalls of dependence on external perception tools. Based on this paradigm, we develop **ViLASR**, a **VI**sion-**LA**nguage model that achieves sophisticated **S**patial **R**easoning through interwoven thinking and visual drawing. To realize this vision, we develop a three-stage training framework (Figure 2). First, we introduce cold-start training with synthetic data to establish basic visual drawing abilities. Second, we design a reflective rejection sampling mechanism that selectively reinforces reasoning paths demonstrating both correct answers and self-correction behaviors, enabling models to revise their visual operations based on intermediate results. Finally, we employ reinforcement learning (RL) with carefully designed rewards that balance answer correctness and reasoning format, incentivizing both accurate spatial understanding and coherent visual thinking processes.

Extensive experiments across five diverse spatial reasoning benchmarks demonstrate the effectiveness of ViLASR, which achieves substantial improvements over strong baselines by 18.4% on average, involving challenging scenarios that require sequential spatial planning (e.g., maze navigation), temporal relationship tracking (e.g., video-based reasoning), and information integration from multiple perspectives. Ablation studies reveal the critical role of each training stage. Particularly, reflective rejection sampling significantly enhances the model's self-correction capabilities, with the frequency of reflection behaviors doubling compared to models trained without this intermediate stage. Further inference-time scaling experiments reveal that each training stage progressively enhances the model's reasoning potential, with the final RL optimization effectively consolidating multi-attempt capabilities while maintaining strong single-attempt performance, significantly narrowing the performance gap compared to earlier training stages.

In summary, our contributions are threefold:

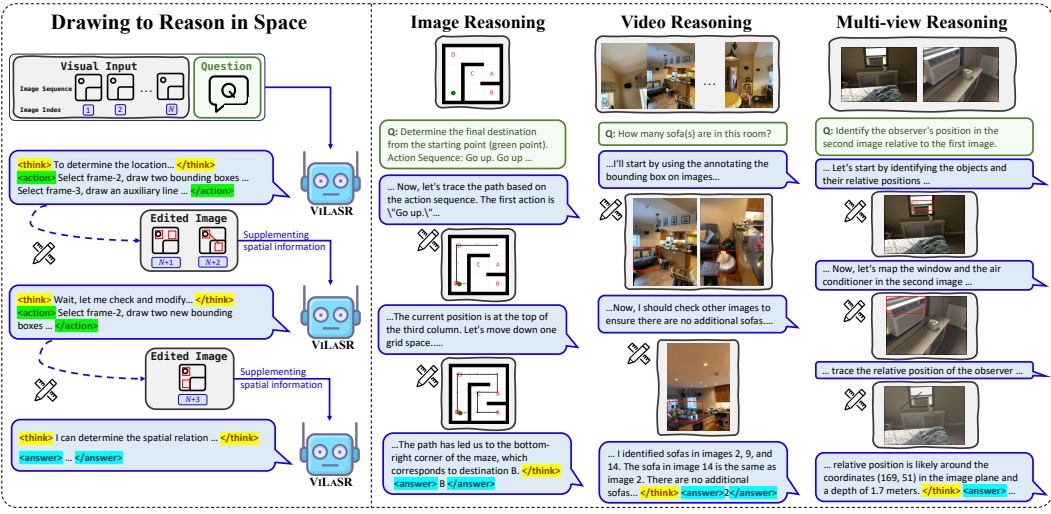

Figure 1: **Top left:** Overview of the "drawing to reason in space" paradigm, which enables visual reasoning through iterative thinking and drawing operations. **Top right:** Examples across three spatial reasoning tasks, demonstrating how ViLASR decomposes complex problems into interpretable visual reasoning steps. **Bottom:** Corresponding results from GPT-4o with only textual reasoning.

I. We propose drawing to reason in space, a novel reasoning paradigm that enables LVLMs to perform spatial reasoning through interpretable visual operations;

II. We develop a principled training framework that effectively cultivates models' visual reasoning capabilities through progressive stages;

III. We demonstrate state-of-the-art performance across multiple spatial reasoning benchmarks and provide insights into the development of visual reasoning abilities in LVLMs.

## 2 Related works

### 2.1 Reasoning in LLMs and LVLMs

Instilling advanced reasoning capabilities within LLMs and LVLMs remains a significant challenge. Current approaches can be broadly categorized into three directions: **(1) Prompt engineering:** crafting prompts to elicit latent reasoning capabilities in pre-trained models [66, 81] and enable tool usage to complement model capabilities with external knowledge [1, 18] or specialized functionalities [57, 19, 61, 73]. These prompt-based approaches heavily depend on models' instruction-following capabilities and often suffer from prompt sensitivity. **(2) Supervised fine-tuning:** developing tailored datasets to enhance specific reasoning capabilities [75, 67, 82] and training models to effectively utilize external tools [54, 53, 18, 8, 55, 51]. Such approaches are inherently bounded by the quality and scale of training data, limiting their potential for advanced reasoning. **(3) Reinforcement learning:** engineering reward mechanisms to incentivize desired reasoning patterns [29, 56, 10] and optimize tool usage strategies [34]. While promising in textual domains, RL approaches for tool-augmented reasoning have yet to be fully investigated in multimodal scenarios. Current LVLMs either confine their reasoning to textual form [72], or rely on specialized perception tools, being constrained by tool capabilities [8, 51] and lacking reflective reasoning [55]. In contrast, we pioneer a staged training recipe that enables models to both express and reflect upon their spatial understanding through iterative drawing operations.

## 2.2 Visual spatial reasoning

Visual spatial reasoning, as a crucial component of multimodal intelligence, extends beyond general visual perception to encompass two particularly challenging fundamental capabilities [15, 77]: relational reasoning, which involves understanding distances, directions, and spatial common sense between objects [38, 60]; and perspective transformation, which requires holding and manipulating spatial relationships [42, 70]. Despite the remarkable progress of current LVLMs in basic visual tasks [24, 36, 39, 9], numerous benchmarks have revealed significant challenges in spatial reasoning [43, 27, 69, 32, 70]. This limitation is partly attributed to the nature of current training datasets [22], which primarily focus on visual perception rather than spatial understanding. Recent works have attempted to address this challenge through various approaches. For image-based spatial reasoning, several studies have proposed synthetic datasets [38, 5, 7, 4]. For video-based spatial reasoning, approaches have emerged either by incorporating 3D representations as bridging knowledge [83] or by tracking objects across frames [37]. However, systematic approaches to enhance LVLMs' spatial reasoning capabilities that can generalize across both images and videos remain largely unexplored. Our work fills this gap by developing a principled reasoning framework in LVLMs.

## 3 Methodology

In this section, we present our approach to advancing spatial reasoning capabilities in LVLMs. Given a spatial reasoning question $Q$ and a visual input $I = \{I_n\}_{n=1}^N$, where each $I_n$ represents a single image, $N = 1$ corresponds to a single image input, and $N > 1$ denotes a video sequence or multiple image inputs, our goal is to enable LVLMs to derive the answer $A$ through iterative visual drawing and thinking. We first introduce our reasoning paradigm (3.1), and then detail the training framework (3.2) that cultivates this reasoning capability.

### 3.1 Drawing to reason in space

We propose "drawing to reason in space," a reasoning paradigm that empowers LVLMs to decompose complex spatial reasoning tasks into a sequence of interpretable visual drawing and thinking steps, as illustrated in Figure 1. Formally, given the visual input $I$ and question $Q$, the LVLM $\mathcal{M}$ generates a multi-step reasoning path $R$ to derive the final answer. The reasoning path $R$ can be represented as a $T$-step chain: $R = \{(r_t, e_t, o_t)\}_{t=1}^T$, where each step interleaves natural language reasoning $r_t$, drawing operations $e_t$, and observed results $o_t$ from executing the drawing operations [74]. This iterative process continues until $\mathcal{M}$ reaches a conclusive answer in the final reasoning step $r_T$. Next, we elaborate on the core components of the paradigm.

**Drawing operations for spatial reasoning.** We equip LVLMs with two essential drawing operations $\mathcal{T} = \{\mathcal{T}_{\text{box}}, \mathcal{T}_{\text{line}}\}$ for bounding box annotation and auxiliary line drawing, respectively. These operations are fundamental to spatial reasoning as they enable explicit representation of object locations and their spatial transitions—bounding boxes anchor object positions while auxiliary lines visualize spatial trajectories and relationships[2]. Each operation $\tau \in \mathcal{T}$ accepts three parameters:

- $k$ : the index of the target image from either the original input $I$ or previous drawing outputs $\{o_i\}$;
- $p$ : single or multiple spatial coordinates for annotating bounding boxes or drawing auxiliary lines;
- $l$ : semantic labels describing the annotated content.

The execution output of each drawing operation is an annotated version of the target image, preserving the original content while overlaying the specified visual elements. While this minimal operation set proves effective for spatial reasoning, it can be readily extended with additional visual manipulation tools in future work.

**Per-step spatial reasoning.** At the $t$-th reasoning step, $\mathcal{M}$ generates both natural language reasoning $r_t$ and drawing operations $e_t$ based on the entire interaction history:

$$\left(r_t, e_t = \{e_t^j = (k_t^j, p_t^j, l_t^j)\}_{j=1}^{m_t}\right) = \mathcal{M}(I, Q, R_{<t}), \tag{1}$$

---

[2]Since this work focuses on inter-object spatial relationships rather than object-specific details, we adopt only basic drawing operations rather than common visual manipulations like zooming or cropping [8, 47].

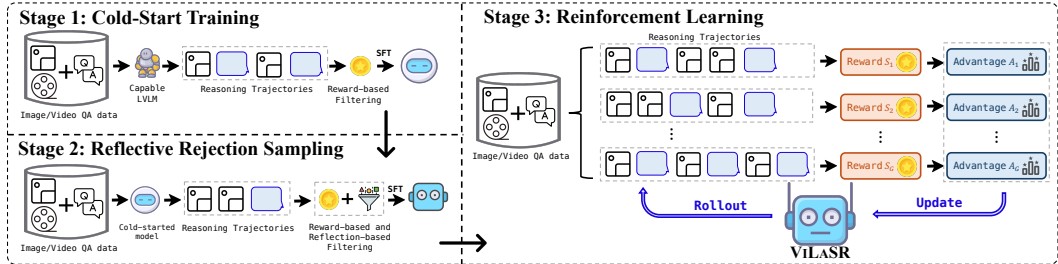

Figure 2: Overview of the three-stage training framework of ViLaSR.

where $e_t$ specifies a set of $m_t$ operations to be executed, $e_t^j$ denote each single operation with three necessary parameters, and $R_{<t} = \{(r_i, e_i, o_i)\}_{i<t}$ means the reasoning history before step $t$. The operations in $e_t$ can target different images through distinct indices $k_t^j$, which is crucial for both image and video reasoning as key information may come from multiple previous drawing outputs or different video frames. After execution, these operations output a set of annotated images $o_t = \{o_t^j\}_{j=1}^{m_t}$, which are sequentially indexed to maintain order. To enable index-based image retrieval for drawing, $\mathcal{M}$ has access to the indices of all available images, including both the original visual input and operation-generated outputs. The reasoning process terminates under two conditions: (i) when $r_t$ reaches a final answer, in which case $e_t = \emptyset$ and $o_t = \emptyset$; or (ii) when the number of reasoning steps reaches a predefined maximum $T_{max}$.

## 3.2 Training framework

Our training framework consists of three stages: cold-start training with synthetic offline data to establish basic visual interaction capabilities, reflective rejection sampling to cultivate reflection behaviors, and reinforcement learning for incentivizing reasoning potentials, as illustrated in Figure 2. During training, we focus on multiple-choice questions (e.g., "Which object is to the left of the chair? A. table ...") and numerical questions (e.g., "How many tables appear in this video?") due to their amenability to automated correctness evaluation, strategically excluding free-form answer questions (e.g., describing motion trajectories). In what follows, we first present our reward function design that guides all three stages, followed by detailed descriptions of the stage-wise training procedures.

**Reward function.** For reward design, we propose a rule-based function that combines answer correctness and reasoning format adherence:

$$S = \mathbb{1}\big(S_{\text{correct}} > \beta\big) \cdot \big(S_{\text{correct}}(A, \hat{A}) + S_{\text{format}}(R)\big), \tag{2}$$

where $\beta$ is a threshold that ensures format rewards are only granted when a minimum correctness threshold is met, preventing reward hacking where the model might optimize for format adherence at the expense of task accuracy. $\hat{A}$ is the predicted answer that we extract from the model's final reasoning step $r_T$ using rule-based parsing. Specifically, we compute $S_{\text{correct}}$ as follows:

$$S_{\text{correct}}(A, \hat{A}) = \begin{cases} \mathbb{1}(A = \hat{A}), & \text{for multiple-choice questions,} \\ \frac{1}{|\mathcal{C}|} \sum_{\theta \in \mathcal{C}} \mathbb{1}\left(\frac{|A - \hat{A}|}{A} < 1 - \theta\right), & \text{for numerical questions.} \end{cases} \tag{3}$$

For multiple-choice questions, we simply check the exact match between the predicted and ground-truth answers. For numerical questions, we adopt Mean Relative Accuracy (MRA) [70] for reward computation, which provides a more robust evaluation than conventional metrics like absolute error or fixed thresholding. Instead, MRA examines the prediction accuracy across multiple confidence levels $\mathcal{C} = \{0.50, 0.55, \dots, 0.95\}$. For each threshold $\theta \in \mathcal{C}$, it checks if the relative error between predicted value $A$ and ground truth $\hat{A}$ falls below $(1 - \theta)$. The final score averages these binary outcomes, effectively measuring the model's precision across different stringency levels.

For $S_{\text{format}}$, we evaluate the quality of reasoning format based on the structural validity of $R$, assigning a score of 1 if all operations in the reasoning path $R$ are executable, and 0 otherwise.

**Cold-start training.** Prompting alone often fails to elicit effective visual manipulation abilities in LVLMs for reasoning [19, 8]. Therefore, we initialize models' ability to reason in space with

drawing operations through supervised learning on a synthetic dataset $\mathcal{D}_{\text{cold}}$. The training objective is to minimize the average negative log-likelihood over all reasoning and operation tokens:

$$\mathcal{L}_{\text{cold}} = \mathbb{E}_{(I,Q,A,R=\{(r_t,e_t,o_t)\}_{t=1}^T)\sim\mathcal{D}_{\text{cold}}}\Big(-\frac{1}{N}\sum_{t=1}^T \log p(r_t,e_t|I,Q,R_{<t})\Big) \tag{4}$$

where $N = \sum_{t=1}^T(|r_t| + |e_t|)$ denotes the total number of tokens in reasoning steps and drawing operations. To construct $\mathcal{D}_{\text{cold}}$, we first collect a diverse set of image and video question-answering pairs from publicly available datasets, comprising visual inputs $I$, questions $Q$, and ground-truth answers $A$. We then leverage Qwen2.5-72B-VL [2] to generate reasoning paths following our reasoning paradigm described in §3.1. The generated paths are subsequently filtered based on rule-based correctness and format checking in Eq. 2 to ensure high-quality demonstrations of spatial reasoning. Appendix B shows more details.

**Reflective rejection sampling.** The success of RL often relies on the model's initial capability to exhibit reflective behaviors [14]. In our case, the ability to reflect on and revise drawing operations based on observed execution output is crucial. Accordingly, we define reflective behavior as the recurrence of identical labels across different time steps within the same reasoning process. Formally, for a reasoning path $R$, reflection occurs when:

$$\exists(t_1,t_2,u,v): (l_{t_1}^u = l_{t_2}^v) \wedge (e_{t_1}^u \neq e_{t_2}^v) \text{ in } R = \{(r_t, e_t = \{e_t^j = (k_t^j, p_t^j, l_t^j)\}_{j=1}^{m_t}, o_t)\}_{t=1}^T, \tag{5}$$

where $l_{t_1}^i$ and $l_{t_2}^j$ represent semantic labels assigned to drawing operations at different reasoning steps, and $e_t^j$ denote the parameters of one drawing operation. However, after cold-start training, we observe that the resulting model, denoted as $\mathcal{M}_{\text{cold}}$, infrequently exhibits such reflective behavior (see §4.3 for more details), potentially limiting the effectiveness of subsequent RL optimization. To address this limitation, we introduce a novel reflective rejection sampling mechanism. Given a batch of spatial reasoning examples $\mathcal{D}_{\text{reflect}}$, we first use $\mathcal{M}_{\text{cold}}$ to sample corresponding reasoning paths, and then continue fine-tune $\mathcal{M}_{\text{cold}}$ with the following objective:

$$\mathcal{L}_{\text{reflect}} = \mathbb{E}_{(I,Q,A)\sim\mathcal{D}_{\text{reflect}},R\sim\mathcal{M}_{\text{cold}}(\cdot|I,Q)}\Big(-\frac{1}{N}\sum_{t=1}^T \log p(r_t,e_t|I,Q,R_{<t})\Big)\phi, \tag{6}$$

where $\phi$ acts as a binary filter that equals 1 if and only if the reasoning path $R$ not only yields the correct answer $A$ and meets format criteria, but also satisfies the reflection condition in Eq. 5. This selective training strategy encourages the model to learn from high-quality reasoning paths that demonstrate both reflective thinking and correct reasoning, facilitating the development of self-correction capabilities crucial for subsequent RL optimization.

**Reinforcement learning.** We further optimize the model using RL with carefully designed rollout policies. During policy rollout, we monitor the reasoning process and apply early termination to avoid inefficient or circular reasoning patterns. Specifically, the rollout is terminated and the reward is set to zero when any of the following conditions are met: (1) the model fails to generate any drawing operations, i.e., $e_t = \emptyset$ while $r_t$ has not reached a final answer; (2) the number of accumulated images exceeds a predefined threshold $\alpha$; or (3) a drawing operation duplicates a previously executed one, i.e., $\exists t_1, t_2, u, v : e_{t_1}^u = e_{t_2}^v$.

With the rollout policies defined above and reward functions in Eq. 2, we optimize the policy using GRPO [10] without the KL penalty term [20]:

$$\mathcal{L}_{\text{RL}} = \mathbb{E}_{\substack{(I,Q,A)\sim\mathcal{D}_{\text{rl}}\\\{R_i\}_{i=1}^G\sim p_{\text{old}}(\cdot|I,Q)}}\left(-\frac{1}{G}\sum_{i=1}^G\frac{1}{N_i}\sum_{t=1}^{T_i}\min\left(\rho_{i,t}A_i,\,\text{clip}(\rho_{i,t},1-\epsilon,1+\epsilon)A_i\right)\right), \tag{7}$$

$$\rho_{i,t} = \frac{p(r_{i,t},e_{i,t}|I,Q,R_{i,<t})}{p_{\text{old}}(r_{i,t},e_{i,t}|I,Q,R_{i,<t})}, A_i = \frac{S_i - \text{mean}(\{S_j\}_{j=1}^G)}{\text{std}(\{S_j\}_{j=1}^G)}, \tag{8}$$

where $G$ is the number of rollout reasoning paths, $R_i = \{(r_{i,t}, e_{i,t}, o_{i,t})\}_{t=1}^{T_i}$ is the $i$-th reasoning path, $N_i$ denotes the total length of the $R_i$ except the tool outputs, $S_i$ is the reward of $R_i$, and $p$ and $p_{\text{old}}$ represent the current and old policy distributions, respectively. The normalized score $A_{i,t}$ reflects the relative quality of each reasoning path within the rollout group, enabling the model to distinguish between learnable and poor reasoning trajectories.

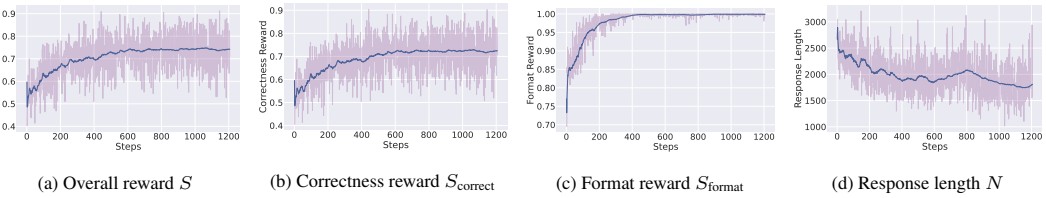

| (a) Overall reward $S$ | (b) Correctness reward $S_{correct}$ | (c) Format reward $S_{format}$ | (d) Response length $N$ |

Figure 3: RL training curves of VILASR.

## 4 Experiment

### 4.1 Experimental setups

**Evaluation benchmark and metrics.** To evaluate the effectiveness and generalization of VILASR, we conduct experiments on five benchmarks covering three categories:

- **Image spatial reasoning**: focusing on static relationships and sequential planning, including (1) Maze [25], specifically designed for navigation assessment; (2) SpatialEval-Real [65], demanding real-world spatial relation understanding.
- **Video spatial reasoning**: requiring temporal relationship tracking, including VSI-Bench [70], which tests visual spatial understanding over temporal sequences.
- **Multi-view spatial reasoning**: challenging models to integrate information from multiple perspectives, including SPAR-Bench [78], MMSI-Bench [71].

These benchmarks consist exclusively of multiple-choice and numerical questions. We evaluate the model performance using accuracy for multiple-choice questions and Mean Relative Accuracy (MRA) [70] for numerical questions. Notably, these metrics follow the same formulation as our training objective (Eq. 3). Appendix C.2 shows the benchmark statistics.

**Implementation details.** We implement VILASR based on Qwen2.5-VL-7B [2]. During the training phase, we process all visual inputs at a maximum resolution of $256 \times 28 \times 28$, with video clips uniformly sampled at 16 frames. In the evaluation stage, we maintain the 16-frame count and the $256 \times 28 \times 28$ frame resolution for VSI-Bench, while increase the resolution to $448 \times 28 \times 28$ for other benchmarks. The training is conducted on a cluster of 16 NVIDIA A100 (80G) GPUs. For both cold-start training stage and reflective fine-tuning stage, we optimize the model with a learning rate of $1 \times 10^{-5}$ for three epochs, requiring approximately 24 and 3 hours, respectively. The subsequent RL optimization is implemented using the VERL framework [58], where we set the training batch size to 32 and generate 8 candidate reasoning paths per question. We set the maximum cumulative image number $\alpha$ to 42. And the reward threshold $\beta$ in Eq. 2 is set to 0.0.

The training dynamics are shown in Figure 3. As illustrated in (a-c), both overall reward $S$ and its components ($S_{correct}$ and $S_{format}$) demonstrate steady improvements, validating the effectiveness of our RL optimization. Notably, while achieving better task performance, we observe a decrease in response length $N$ from 2,500 to 1,800 tokens (d). This can be attributed to two factors: First, during cold-start training, we set a minimum of three reasoning steps in synthetic data to encourage in-depth reasoning, while RL optimization subsequently drives the model toward more efficient use of drawing operations. Second, the limited training data may not fully expose the model to complex scenarios requiring lengthy reasoning chains. We expect that increasing training data diversity could potentially lead to longer but necessary reasoning steps for more challenging spatial tasks [68]. We also present an analysis of the runtime performance in Appendix E.2.

**Baselines.** We compare VILASR with various representative models and methods as follows:

- **Proprietary LVLMs**: GPT-4o [44], Gemini-1.5-Pro [62], Gemini-2.0-Flash [16], OpenAI o3 and o4-mini [46];
- **Open-source LVLMs**: These models range from small-sized models including Qwen2.5-VL-7B [2], LLaVA-NeXT-Video-7B [80] and LLaVA-OneVision-7B [30] to large-sized models including Kimi-VL-A3B-Instruct (16B) [63], Qwen2.5-VL-72B [2], LLaVA-NeXT-Video-72B [80], LLaVA-OneVision-72B [30];

Table 1: Performance comparison across spatial reasoning benchmarks. Gray-shaded rows represent large-sized models (>7B parameters). For non-shaded rows, **bold** and underlined numbers indicate the best and second-best results. *Italic* numbers in gray-shaded rows indicate performance below non-shaded best results. *Improvement* refers to the absolute improvement of VɪLaSR compared with Qwen2.5-VL-7B w/o reasoning. † and ‡ indicate results from VSI-Bench and VSI-Bench (tiny) set [70] respectively. ⋆ Results from [63]. ⋆⋆ Results from [71]. N/A: not support multiple-image input. OpenAI o4-mini is evaluated on a small subset of benchmarks due to its high cost.

| Method | Tool | Reasoning | Image | | Video | Multi-view | |
| --- | --- | --- | --- | --- | --- | --- | --- |
| | | | MAZE | SpatialEval-Real | VSI-Bench | SPAR-Bench | MMSI-Bench |
| *Proprietary LVLMs* | | | | | | | |
| GPT-4o | ✗ | ✗ | *48.8* | *60.7* | *34.0†* | *33.6* | 30.3⋆⋆ |
| GPT-4o | ✗ | ☑ | *63.1* | 65.1 | - | 38.1 | - |
| Gemini-1.5-Pro | ✗ | ✗ | - | - | *45.4†* | - | - |
| Gemini-2.0-Flash | ✗ | ✗ | *40.2* | - | *45.4‡* | *33.4* | - |
| Gemini-2.0-Flash | ✗ | ☑ | *61.7* | - | - | *28.0* | - |
| OpenAI o3 | ✗ | ☑ | - | - | - | - | 41.0⋆⋆ |
| OpenAI o4-mini | ✗ | ☑ | *79.0* | - | - | 46.2 | - |
| *Open-source LVLMs* | | | | | | | |
| Qwen2.5-VL-7B | ✗ | ✗ | 33.7 | 58.5 | 32.7 | 31.7 | 26.9 |
| Qwen2.5-VL-7B | ✗ | ☑ | 36.5 | 54.1 | 26.2 | 31.6 | 27.1 |
| LLaVA-NeXT-Video-7B | ✗ | ✗ | 34.7 | **68.1** | 35.6† | 31.3 | 26.8 |
| LLaVA-OneVision-7B | ✗ | ✗ | 30.8 | 62.9 | 32.4† | 30.6 | 24.5⋆⋆ |
| Kimi-VL-A3B-Instruct (16B) | ✗ | ✗ | 45.0 | 68.9 | *37.4⋆* | *33.0* | - |
| Qwen2.5-VL-72B | ✗ | ✗ | *50.5* | 68.1 | 36.0 | 37.4 | 30.7⋆⋆ |
| LLaVA-NeXT-Video-72B | ✗ | ✗ | *43.2* | 72.5 | *40.9†* | *35.6* | 28.3 |
| LLaVA-OneVision-72B | ✗ | ✗ | *46.3* | 68.8 | *40.2†* | *34.4* | 28.4⋆⋆ |
| *Representative methods for multimodal reasoning* | | | | | | | |
| CoGCoM-17B [51] | ☑ | ☑ | *29.8* | *49.3* | *N/A* | *N/A* | *N/A* |
| VisCoT-7B [55] | ☑ | ☑ | 26.0 | 43.2 | N/A | N/A | N/A |
| SpaceR-7B [48] | ✗ | ☑ | 38.6 | 62.7 | 43.5 | 37.1 | 28.8 |
| VisProg [19] | ☑ | ☑ | 22.6 | 26.8 | N/A | N/A | N/A |
| ViperGPT [61] | ☑ | ☑ | 27.3 | 45.9 | 10.6 | 23.7 | 25.4 |
| *Ours* | | | | | | | |
| VɪLaSR | ☑ | ☑ | **98.2** | 63.9 | **45.4** | **37.6** | **30.2** |
| *Improvement* | *N/A* | *N/A* | *+64.5* | *+5.4* | *+12.7* | *+5.9* | *+3.3* |

- **Representative models focused on multimodal reasoning**: CoGCoM [51], which enables step-by-step visual reasoning through image manipulations with specialized perception tools (e.g., grounding) but is limited to single-image inputs; VisCoT [55], which first identifies key regions through bounding box annotations followed by textual reasoning chains on single images; SpaceR [48], which is a contemporary work that extends spatial reasoning to video understanding through task-specific optimization on automatically curated video QA data; VisProg [19], which pioneers visual programming by prompting an LLM to generate pseudocode orchestrating modular vision APIs for compositional tasks; and ViperGPT [61], which extends this to generate executable Python with richer control flow (e.g., loops, conditionals) for vision–language interaction.

Furthermore, we perform ablation studies to evaluate each individual training stage to assess their contributions to the final performance. Implementation details of the baselines are show in Appendix C.1.

## 4.2 Main results

As shown in Table 1, VɪLaSR shows strong performance across various spatial reasoning benchmarks, from maze navigation to complex video understanding. Our analysis reveals several key findings:

**(1) Limited visual reasoning capabilities in open-source LLMs.** Our experiments reveal a significant disparity between proprietary and open-source LLMs in their ability to leverage reasoning processes. Proprietary models consistently outperform their non-reasoning counterparts by a substantial margin across different benchmarks (e.g., GPT-4o, Gemini-2.0-Flash). However, open-source models like Qwen2.5-VL-7B show minimal or even negative improvements with reasoning enabled, suggesting a significant gap in their fundamental visual reasoning capabilities. This observation highlights a critical direction for future research: enhancing the multimodal reasoning abilities of

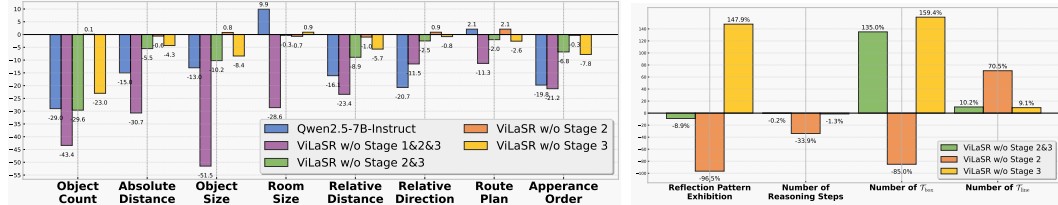

(a) Absolute change in scores compared with VᵢLaSR on eight subsets of VSI-Bench.

(b) Relative change (%) in four key behavioral metrics compared with VᵢLaSR.

Figure 4: Ablation study results on VSI-Bench regarding three training stages: cold-start training (Stage 1), reflective rejection sampling (Stage 2), and reinforcement learning (Stage 3).

open-source models, potentially through improved architectural designs and more sophisticated training strategies that better integrate visual and linguistic information.

**(2) Strong performance of VᵢLaSR on image-based spatial tasks.** VᵢLaSR demonstrates exceptional capabilities in both sequential spatial planning (MAZE) and static spatial understanding (SpatialEval). The remarkable performance on maze navigation, in particular, highlights the advantage of our "drawing to reason in space" paradigm: through iterative drawing operations, the model effectively breaks down multi-step navigation sequences into interpretable visual steps, thereby capturing and tracking spatial state transitions. In contrast, existing methods show limited performance due to various constraints: CᴏɢCᴏM is restricted by the capabilities of its perception tools, and VisCoT lacks the ability to reflect and revise its visual operations since it only performs one-time grounding. VisProg and ViperGPT exhibit inferior performance because they often generate malformed or non-executable programs on complex questions. Moreover, despite leveraging modular APIs, these methods frequently struggle with basic spatial relationships, consistent with recent findings [41].

**(3) Competitive results of VᵢLaSR on video and multi-view reasoning.** Video and multi-view reasoning present unique challenges beyond static image understanding, as it requires tracking spatial relationships across multiple viewpoints and temporal transitions. While most existing models struggle with this increased complexity, VᵢLaSR achieves state-of-the-art performance on all benchmarks, surpassing significantly larger open-source models such as LLaVA-OneVision-72B. This success can be attributed to our flexible visual operations that effectively handle dynamic spatial relationships. In contrast, existing approaches face fundamental limitations: CᴏɢCᴏM and VisCoT lack video processing capabilities entirely, while SpaceR's text-centric reasoning fails to fully exploit visual temporal information.

## 4.3 Ablation study

To comprehensively evaluate our training framework, we conduct ablation studies on VSI-Bench's eight spatial reasoning subtasks and analyze four key behavioral metrics. The behavioral metrics include: (1) *Reflection Pattern Exhibition*, measuring the ratio of examples exhibiting reflection behavior as defined in Eq. 5; (2) *Number of Reasoning Steps*, indicating the average number of reasoning steps ($T$) required to answer each question; and (3,4) *Numbers of $\mathcal{T}_{\text{box}}$ and $\mathcal{T}_{\text{line}}$*, reflecting the average number of drawing operations utilized per question. The results are shown in Figure 4(a) for task performance and Figure 4(b) for behavioral patterns. We draw the following insights:

**(1) Cold-start training: Essential for spatial reasoning foundation.** Comparing "VᵢLaSR w/o Stage 1&2&3" with "VᵢLaSR w/o Stage 2&3", we observe substantial performance improvements across all subtasks. Notably, when equipped with only our "drawing to reason in space" paradigm without training (i.e., "VᵢLaSR w/o Stage 1&2&3"), the model performs even worse than the Qwen2.5-VL-7B backbone. This degradation highlights that sophisticated spatial reasoning capabilities cannot emerge solely with prompting but must be learned through dedicated training.

**(2) Reflection sampling: Key to self-correction ability.** The comparison between VᵢLaSR and "VᵢLaSR w/o Stage 2" reveals the crucial role of reflective rejection sampling. Removing this stage leads to a 96.5% decrease in reflection pattern exhibition, fundamentally altering the model's behavior and performance. Behaviorally, we observe a 33.9% reduction in reasoning steps and dramatically different patterns in drawing operation usage: a 85.0% decrease in $\mathcal{T}_{\text{box}}$ but a 70.5%

increase in $\mathcal{T}_{\text{line}}$. Such changes particularly affect tasks requiring precise object localization and measurement: "Absolute Distance" (-0.6%), "Room Size" (-0.7%), and "Relative Distance" (-1.0%). In contrast, tasks that primarily rely on directional judgment or categorical reasoning show minimal impact or even slight improvements (e.g., "Object Count": +0.1%, "Relative Direction": +0.9%), as they can be solved with simpler spatial relationships indicated by lines. This pattern reveals that reflection capability fundamentally shapes how the model builds spatial understanding. Without it, the model shows a tendency to make quick spatial judgments through auxiliary lines without sufficient self-verification. Based on this observation, we hypothesize specific mechanisms through which reflection enhances spatial reasoning: re-examining object locations through bbox annotations and carefully evaluating different spatial relationships before making final judgments (see examples in Appendix §D). This verification-driven approach, rather than making multiple "guessing" attempts through auxiliary lines, appears to be key to accurate spatial reasoning.

**(3) RL optimization: Dense rewards enable fine-grained learning.** Comparing VILASR with "VILASR w/o Stage 3", we observe consistent performance decreases across most subtasks. The substantial increases in both $\mathcal{T}_{\text{box}}$ (+159.4%) and $\mathcal{T}_{\text{line}}$ (+9.1%) usage without RL suggest that RL optimization helps the model learn to use drawing operations more selectively. Moreover, tasks requiring precise numerical answers ("Object Count," "Absolute Distance," "Object Size" and "Room Size") show greater average performance gaps without RL compared to multiple-choice questions (-9.21% vs. -4.07%). This disparity highlights a key advantage of RL optimization: while supervised learning merely maximizes the probability of correct answers, RL provides dense reward signals based on numerical proximity to ground truth, enabling more effective learning for precise spatial measurements.

### 4.4 Inference-time scaling

We further evaluate the effectiveness of our training framework through the lens of inference-time scaling analysis such as pass@$k$ evaluation (sampling $k$ outputs in parallel and selecting the best). Prior research has demonstrated: (1) pass@$k$ serves as an empirical upper bound of model capability with sufficiently large $k$ [59], while pass@$1$ reflects its single-attempt performance; (2) RL has been shown to effectively narrow this gap by consolidating pass@$k$ capabilities into pass@$1$ performance [76]. This motivates our analysis of how each training stage expands model capability.

Results in Figure 5 reveal the progressive enhancement of model capability across training stages: (1) Cold-start training substantially improves both the empirical upper bound (pass@$8$) and single-attempt performance (pass@$1$) compared to Qwen2.5-VL-7B; (2) Reflective rejection sampling further elevates both metrics, demonstrating expanded reasoning potential; (3) RL optimization not only pushes the empirical upper bound (pass@$8$) but also significantly narrows the gap with single-attempt performance (pass@$1$), indicating effective consolidation of multi-attempt capabilities that aligns with previous studies about RL's optimization effects [76].

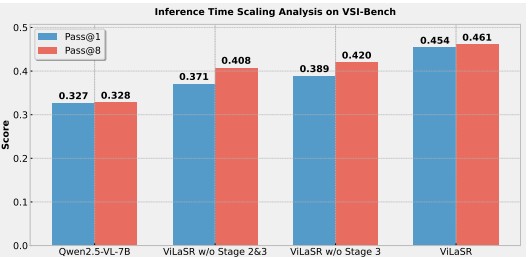

Figure 5: Analysis of inference-time scaling behavior using pass@$1$ and pass@$8$ performance.

## 5   Conclusion

This work presents a principled approach to enhancing spatial reasoning capabilities in LVLMs through visual drawing operations. By enabling direct manipulation in the visual space through the "drawing to reason in space" paradigm, we bridge the gap between text-centric reasoning and human-like spatial cognition. Through careful empirical validation, we demonstrate that our three-stage training framework successfully cultivates sophisticated reasoning pattern. Extensive experiments across diverse spatial reasoning benchmarks validate the effectiveness of our approach, showing particular strengths in maze navigation and temporal-spatial understanding tasks. While our results are promising, several important directions remain for future exploration, such as extending our drawing operations to handle more complex 3D spatial relationships, and investigating more efficient training strategies.

## Acknowledgements

We thank the anonymous reviewers for their insightful comments. This work was supported in part by the National Natural Science Foundation of China (NSFC) under Grant Nos. 62141608, 62236010, 62372454, and 62576339, and in part by the Ant Group Research Intern Program.

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

# A   Appendix outline

In these supplementary materials, we provide:

- Dataset construction (Appendix B);

- Experimental setup and full evaluation results (Appendix C);

- Visualization results (Appendix D).

- Complexity comparison between VILASR and baselines (Appendix E);

- Discussion on limitations (Appendix F) and broader impact (Appendix G) of VILASR.

# B   Dataset construction

We created diverse datasets spanning three categories of spatial reasoning tasks: maze navigation that tests path planning abilities, static image understanding that focuses on spatial relationships, and video comprehension that captures temporal evolution and multi-view spatial reasoning. Building upon basic spatial understanding in static images through bounding box annotation and auxiliary line drawing, our video data further extends these capabilities by incorporating temporal dynamics and multi-view perspectives, leading to more comprehensive spatial reasoning abilities.

Our data collection pipeline leverages multiple public datasets, with careful balancing across different task types to ensure comprehensive coverage. This effort resulted in three primary datasets: **VI-LASR-ColdStart-33k**, a supervised dataset containing curated reasoning paths, **VILASR-RRS-8k**, specifically designed for the reflective rejection sampling stage, and **VILASR-RL-40k**, optimized for reinforcement learning. For VILASR-ColdStart-33k construction, we leveraged Qwen2.5-VL-72B [2] to generate initial reasoning trajectories following our "drawing to reason in space" paradigm detailed in §3.1. A rigorous filtering process based on answer correctness and format validity helped identify the most reliable samples. We applied two format criteria: (1) ensuring all drawing operations are executable and the final answer can be correctly parsed, and (2) requiring at least three rounds of thinking to ensure sufficient spatial reasoning depth. After cold-start training, we constructed VILASR-RRS-8k by selecting instances that are generated by the cold-started model and exhibit clear self-correction patterns and correct final answers, providing ideal training examples for cultivating reflective behaviors.

Table 2: Dataset distribution across different training stages.

| Subset (Modality) | VILASR-ColdStart-33k | VILASR-RRS-8k | VILASR-RL-40k |
|---|---|---|---|
| Maze Navigation (Image) | 7,000 | 1,000 | 10,000 |
| VQA (Image) | 10,000 | 3,000 | 10,000 |
| GPT4Scene [52] (Video) | 6,000 | 2,000 | 10,000 |
| SR-91k [48] (Video) | 10,000 | 1,800 | 10,000 |
| **Total** | 33,000 | 7,800 | 40,000 |

Table 2 presents the detailed distribution of these datasets. Our source data encompasses a broad range of spatial reasoning scenarios, drawing from GQA [22], VSR [79], OpenImages [28], OpenSpaces [5] and SpaceLLaVA [5]. For maze navigation tasks, we procedurally generated mazes with varying grid sizes (from $3\times3$ to $6\times6$) using a depth-first search algorithm[3], maintaining approximately equal proportions across different grid sizes. Each maze consists of a clearly marked starting point and four candidate destinations (labeled as A, B, C, and D). The questions are formulated as multiple-choice problems where the model must determine which destination would be reached by following a given action sequence (e.g., "Determine the final destination from the starting point (green point). Action Sequence: Go up. Go up ..."). This automated generation process ensures diverse maze layouts and action sequences while maintaining problem solvability, creating a controlled environment for evaluating sequential spatial reasoning capabilities.

---

[3] https://github.com/understanding-search/maze-dataset

Table 3: APIs used for proprietary models evaluation.

| Model | API Name | Provider |
|---|---|---|
| GPT-4o | gpt-4o-2024-08-06 | OpenAI |
| Gemini-1.5-Flash | gemini-1.5-flash | Google |
| Gemini-1.5-Pro | gemini-1.5-pro | Google |
| Gemini-2.0-Flash | gemini-2.0-flash | Google |
| OpenAI o4-mini | o4-mini-2025-04-16 | OpenAI |

Table 4: Number of examples in our evaluation benchmark.

| Image | | Video | Multi-view | |
|---|---|---|---|---|
| MAZE | SpatialEval-Real | VSI-Bench | SPAR-Bench | MMSI-Bench |
| 2,000 | 135 | 5,130 | 7,211 | 1,000 |

# C Experimental setup and full evaluation results

## C.1 Baseline implementation

We evaluate VɪLASR against various state-of-the-art models and methods.

For closed-source LVLMs, we directly query their official APIs as shown in Table 3. We evaluate these models using zero-shot prompting by directly providing the input images and questions. Under the setting without reasoning, we explicitly prompt these models to output answers directly without intermediate reasoning steps, i.e., "Answer with the option's letter from the given choices directly" for multiple-choice questions and "Please answer with a single numerical value (e.g., 42 or 3.14)" for numerical responses.

For open-source LVLMs ranging from 7B parameters (Qwen2.5-VL-7B [2], LLaVA-NeXT-Video-7B [80], LLaVA-OneVision-7B [30]) to 72B parameters (Qwen2.5-VL-72B [2], LLaVA-NeXT-Video-72B [80], LLaVA-OneVision-72B [30]). Kimi-VL-A3B-Instruct(16B) [63] leverages Mixture-of-Experts (MoE) architecture with 16.0B total parameters, while dynamically activating 2.8B parameters during inference. We conduct zero-shot evaluation using their standard prompting formats.

For specialized reasoning models, we evaluate using their officially released checkpoints and prompts: CoGCoM [51] with its built-in perception tools, VisCoT [55] with its bounding box annotation pipeline, and SpaceR [48] with its video understanding capabilities.

## C.2 Benchmark statistics

Table 4 shows the benchmark statistics.

## C.3 Detailed results on VSI-Bench

Table 5 provides detailed results on VSI-Bench. VɪLASR achieves state-of-the-art performance with an average accuracy of 45.4%, outperforming all baseline methods by a significant margin (+12.7%). The improvements are particularly pronounced in tasks requiring precise spatial measurements and object localization: "Object Count" (+29.0%), "Absolute Distance" (+15.0%), and "Object Size" (+13.0%). This aligns with our case study observations (see §D) where VɪLASR demonstrates superior capabilities in systematic measurement and spatial reasoning through drawing operations.

Furthermore, the strong performance in "Relative Distance" (+16.1%) and "Relative Direction" (+20.7%) demonstrates VɪLASR's effectiveness in comparative spatial reasoning. By explicitly drawing auxiliary lines to connect and measure between objects, our model can more accurately assess relative positions and orientations. The significant improvement in "Appearance Order" (+19.8%) further highlights VɪLASR's capability in temporal-spatial reasoning, where systematic annotation of objects across multiple frames helps track and verify their sequential relationships.

Table 5: Results on VSI-Bench. Gray-shaded rows represent large-sized models (>7B parameters). † Results from [70]. ‡ Results from VSI-Bench (tiny) set [70]. ⋆ Results from [63].

| Method | Sub-tasks | | | | | | | | Average |
| | Numerical questions | | | | Multiple-choice questions | | | | |
| | Object Count | Absolute Distance | Object Size | Room Size | Relative Distance | Relative Direction | Route Plan | Apperance Order | |
|---|---|---|---|---|---|---|---|---|---|
| *Proprietary LVLMs* | | | | | | | | | |
| GPT-4o | *46.2* | *5.3* | *43.8* | *38.2* | *37.0* | *41.3* | *31.5* | *28.5* | *34.0†* |
| Gemini-1.5-Flash | 49.8 | *30.8* | 53.5 | 54.4 | *37.7* | *41.0* | *31.5* | 37.8 | *42.1†* |
| Gemini-1.5-Pro | 56.2 | *30.9* | 64.1 | 43.6 | 51.3 | 46.3 | 36.0 | *34.6* | *45.4†* |
| Gemini-2.0-Flash | 52.4 | *30.6* | 66.7 | *31.8* | 56.0 | 46.3 | *24.5* | 55.1 | *45.4‡* |
| *Open-source LVLMs* | | | | | | | | | |
| Qwen2.5-VL-7B | 34.5 | 19.4 | 47.6 | **40.8** | 32.8 | 24.5 | 32.5 | 29.4 | 32.7 |
| LLaVA-NeXT-Video-7B | 48.5 | 14.0 | 47.8 | 24.2 | 43.5 | 42.4 | **34.0** | 30.6 | 35.6† |
| LLaVA-OneVision-7B | 47.7 | 20.2 | 47.4 | 12.3 | 42.5 | 35.2 | 29.4 | 24.4 | 32.4† |
| Kimi-VL-A3B-Instruct-16B | - | - | - | - | - | - | - | - | 37.4⋆ |
| Qwen2.5-VL-72B | *33.9* | *27.2* | *59.3* | *28.5* | *47.2* | *35.3* | *22.2* | *34.5* | *36.0* |
| LLaVA-NeXT-Video-72B | *48.9* | *22.8* | *57.4* | *35.3* | *42.4* | *36.7* | 35.0 | *48.6* | *40.9†* |
| LLaVA-OneVision-72B | *43.5* | *23.9* | *57.6* | *37.5* | *42.5* | *39.9* | *32.5* | *44.6* | *40.2†* |
| *Representative methods for multimodal reasoning* | | | | | | | | | |
| SpaceR-7B | 61.9 | 28.6 | **60.9** | 35.2 | 38.2 | **46.0** | 31.4 | 45.6 | 43.5 |
| *Ours* | | | | | | | | | |
| VILASR | **63.5** | **34.4** | 60.6 | 30.9 | **48.9** | 45.2 | 30.4 | **49.2** | **45.4** |
| *Improvement* | *+29.0* | *+15.0* | *+13.0* | *-9.9* | *+16.1* | *+20.7* | *-2.1* | *+19.8* | *+12.7* |

Interestingly, while showing strong performance in most categories, VILASR exhibits slight decreases in "Room Size" (-9.9%) and "Route Plan" (-2.1%). This suggests a limitation in tasks requiring holistic reasoning: "Room Size" estimation demands understanding the complete room layout from partial views and reasoning about unseen spaces, while "Route Plan" needs global path planning across multiple viewpoints. Unlike localized spatial measurements that can be solved through explicit drawing operations, these tasks require more sophisticated global inference capabilities to integrate information across different views and time points. This reveals potential directions for extending our drawing-based framework: we could introduce specialized drawing tools for layout reconstruction and space completion. Such extensions would be natural additions to our current drawing operation set, enabling more comprehensive spatial reasoning capabilities.

## C.4 Statistical significance analysis

We conduct rigorous statistical significance tests to validate our experimental results. Using paired $t$-tests, we find that VILASR significantly outperforms the base model (Qwen2.5-VL-7B) on MAZE, SpatialEval-Real, VSI-Bench, SPAR-Bench and MMSI-Bench ($p < 0.05$). These results demonstrate the robust and consistent advantages of our approach across different spatial reasoning scenarios.

## C.5 Prompt template

In this section, we present the comprehensive prompt templates utilized in VILASR, including both the system prompt and user prompt, as illustrated in Figure 6 and Figure 7, respectively. We use these prompt templates for both generating reasoning paths in cold start data and inference of VILASR.

## C.6 System prompt of VILASR

The system prompt utilized in the VILASR reasoning framework is presented in Figure 6.

## C.7 Query prompt template of VILASR

The query prompt utilized in the VILASR reasoning framework is presented in Figure 7. Our framework utilizes three distinct types of prompts throughout the reasoning process:

### Available tools:
You can use the following two tools to visualize. After each tool usage, you must wait for and analyze the visualization feedback before proceeding.

1. **Object Mapper**
- Purpose: Identifies and maps key items in the space
- Input format: JSON
```json
[{{
"index": i, # Image index
"bbox_2d": [x1, y1, x2, y2],
"label": "object name/description"
}}]
```
- Output: Generates bounding boxes for visual inspection of the i-th image

2. **Path Tracer**
- Purpose: Plots movement or connections between points
- Input format: JSON
```json
[{{
"index": i, # Image index
"start_point_2d": [x1, y1],
"end_point_2d": [x2, y2],
"label": "trace_description"
}}]
```
- Output: Generates visual paths for verification of the i-th image

### Please NOTE the following reasoning techniques:
1. Initial Analysis
- Break down the spatial problem
- Plan your approach

2. Iterative Reasoning for Each Step
- Choose appropriate tool
- Provide absolute coordinates in JSON format (The top-left corner of the image is (0, 0) and the bottom-right corner is ( {width}, {height}))
- Observe the visualization output
- Reflect on the visualization:
* Is the placement/path accurate?
* Does it align with your reasoning?
* What adjustments are needed?
- Backtrack and Adjust:
* If errors found, backtrack to previous step to modify actions or decisions as needed

Figure 6: System prompt used in VILASR.

- **Initial Query Prompt** (Image/Video Query Prompt): Initiates the reasoning process by establishing the initial visual context and query.
- **Follow-up Query Prompt**: Guides subsequent reasoning steps by incorporating intermediate results and accumulated context.
- **Final Answer Prompt**: Terminates the reasoning process and enforces answer generation when either the maximum number of processed images or reasoning steps is reached.

**Video Query Prompt**

⬚ The index of the given image is 1 (width: **{width}**, height: **{height}**).
…

The index of the given image is **{current_image_idx}** (width: **{width}**, height: **{height}**). Then, you can not invoke the Object Mapper or Path Tracer tool. Please answer the initial question and structure your response as required:

**Output Template**

"Multiple Choice": Answer with the option's letter from the given choices directly.
"Numerical": Please answer the question using a single numerical value.
"Regression": Please answer the question using a single numerical value.

Figure 7: Query prompt and output template used in VILASR.

## D  Visualization results

To illustrate the effectiveness of our approach, we present three representative examples from the MAZE, SPAR-Bench, and VSI-Bench benchmarks in Figures 8, 9, and 10, respectively. Note that the "Path Tracer" tool refers to drawing auxiliary lines, and the "Object Mapper" tool refers to annotating bounding boxes.

In the maze navigation task (Figure 8), while Qwen2.5-VL-7B provides incorrect answers directly and GPT-4o attempts textual reasoning but fails to accurately track spatial transitions, VILASR successfully decomposes the task into interpretable steps. By using the "Path Tracer" tool to visualize and verify each movement through auxiliary lines, our model ensures accurate navigation through the maze, leading to the correct destination.

For multi-view reasoning (Figure 9), VILASR demonstrates sophisticated spatial understanding by first mapping key objects ("window," "air conditioner") in both images using "Object Mapper", then systematically analyzing their relative positions. Through careful verification using "Path Tracer," it correctly determines the observer's position change, while both baseline models struggle with perspective transformations and make incorrect judgments about movement direction.

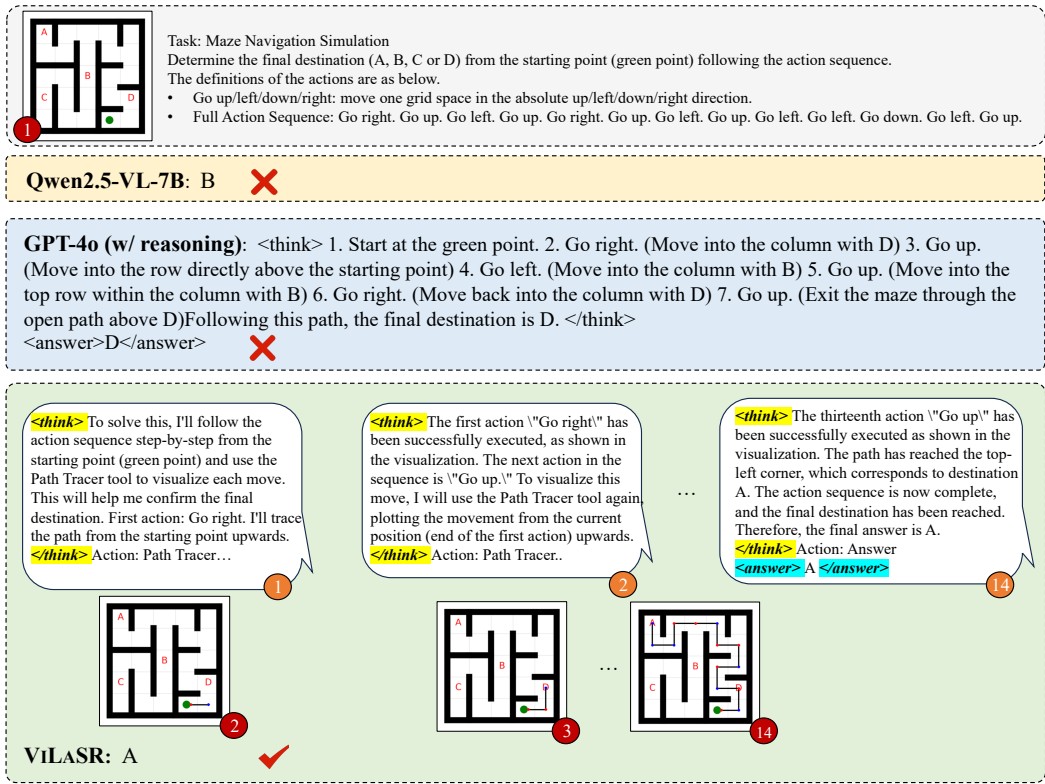

Figure 8: A visualization example of spatial reasoning approaches on the Maze benchmark, including Qwen-2.5-VL-7B, GPT-4o, and VɪLaSR.

In the video spatial reasoning task (Figure 10), VɪLaSR exhibits strong reflection capability and systematic problem-solving. When initial attempts to locate and measure the telephone in images 5 and 12 fail, it self-corrects and identifies the correct object in image 13. Furthermore, it shows a sophisticated measurement strategy by using reference objects ("monitor," "headphones") to establish scale and convert pixel measurements to centimeters. In contrast, baseline models either fail to locate the target object (Qwen2.5-VL-7B) or make rough estimations without proper measurement (GPT-4o).

These cases illustrate how drawing operations enable more reliable spatial reasoning by grounding abstract relationships in concrete visual representations.

# E    Complexity analysis

To thoroughly assess the computational efficiency of VɪLaSR, we present a comprehensive analysis comparing our approach with the base model Qwen2.5-VL-7B model. This analysis encompasses both theoretical complexity bounds and empirical resource utilization, providing insights into the framework's scalability and computational characteristics.

## E.1    Theoretical time complexity

We analyze the computational complexity across three model variants: the base model (performing direct answer generation), standard reasoning models (generating reasoning chains followed by answer derivation), and our VɪLaSR model (employing iterative visual drawing and thinking). Let $M$ denote the input length (including both image and text tokens), $N$ the answer length, $L$ the per-step reasoning path length, and $S$ the number of reasoning steps, where $S = 1$ for both base model and standard reasoning approaches. Empirically, VɪLaSR achieves shorter per-step reasoning

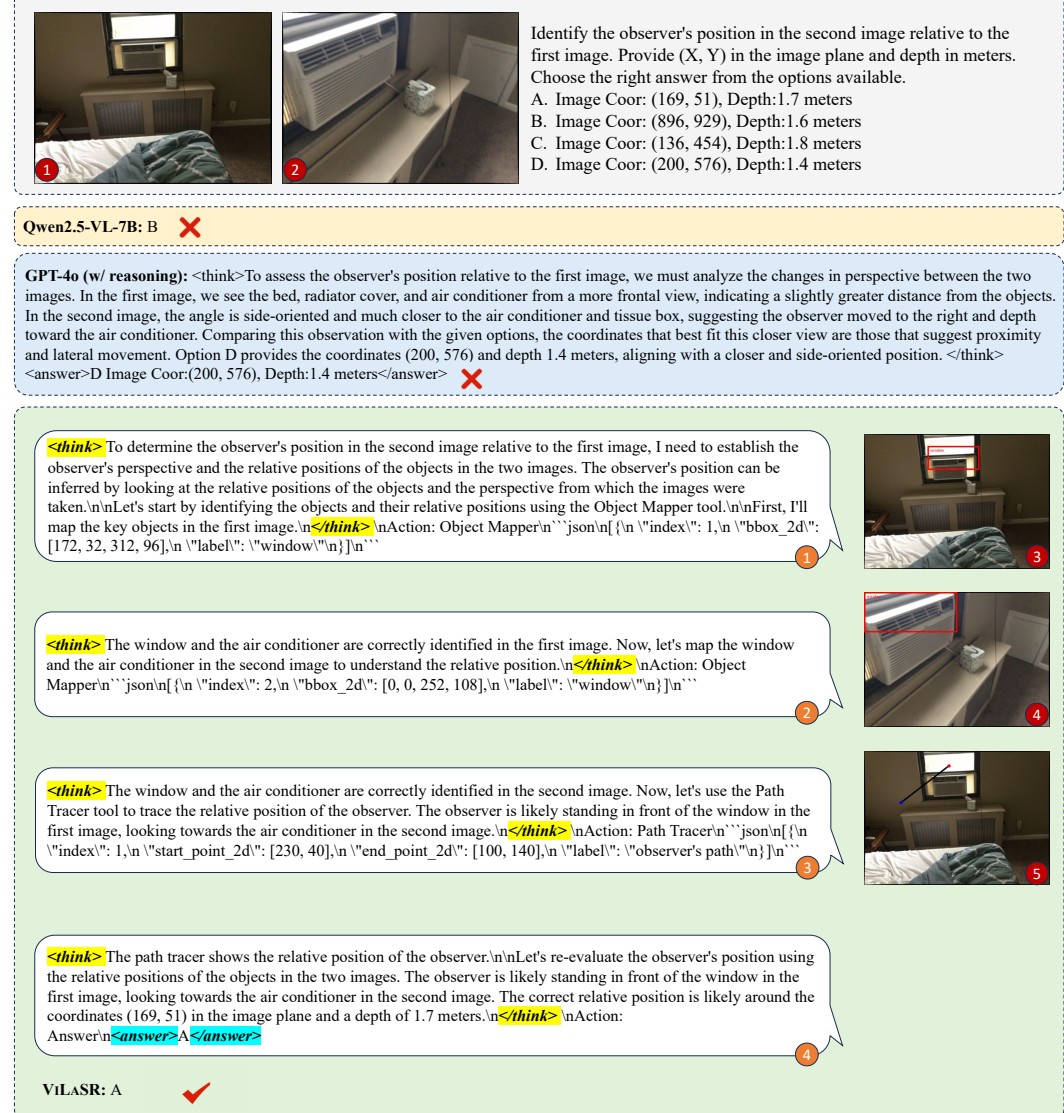

Figure 9: A visualization example of spatial reasoning approaches on the SPAR-Bench benchmark, including Qwen-2.5-VL-7B, GPT-4o, and VɪLaSR.

Table 6: Complexity analysis of different reasoning approaches.

| Model Type | Input Length | Output Length | Time Complexity |
|---|---|---|---|
| Base Model | $M$ | $N$ | $\mathcal{O}(M \cdot N)$ |
| Standard Reasoning | $M$ | $L + N$ | $\mathcal{O}(M \cdot (L + N))$ |
| VɪLaSR (Ours) | $M + \sum_{i=1}^{S} M_i$ | $S \cdot L + N$ | $\mathcal{O}\big(S^2 \cdot L \cdot (L + M) \\ + S \cdot (L + M)(L + N)\big)$ |

paths (smaller $L$) compared to single-step approaches due to its decomposition of reasoning into interpretable visual operations. Table 6 presents a comparative analysis of their time complexities.

The total computational complexity is:

$$T = \sum_{k=1}^{S-1}\Big[\big(M + \sum_{i=1}^{k-1}(L_i + M_i)\big) \cdot L_k\Big] + \big(M + \sum_{i=1}^{S-1}(L_i + M_i)\big) \cdot (L_S + N), \tag{9}$$

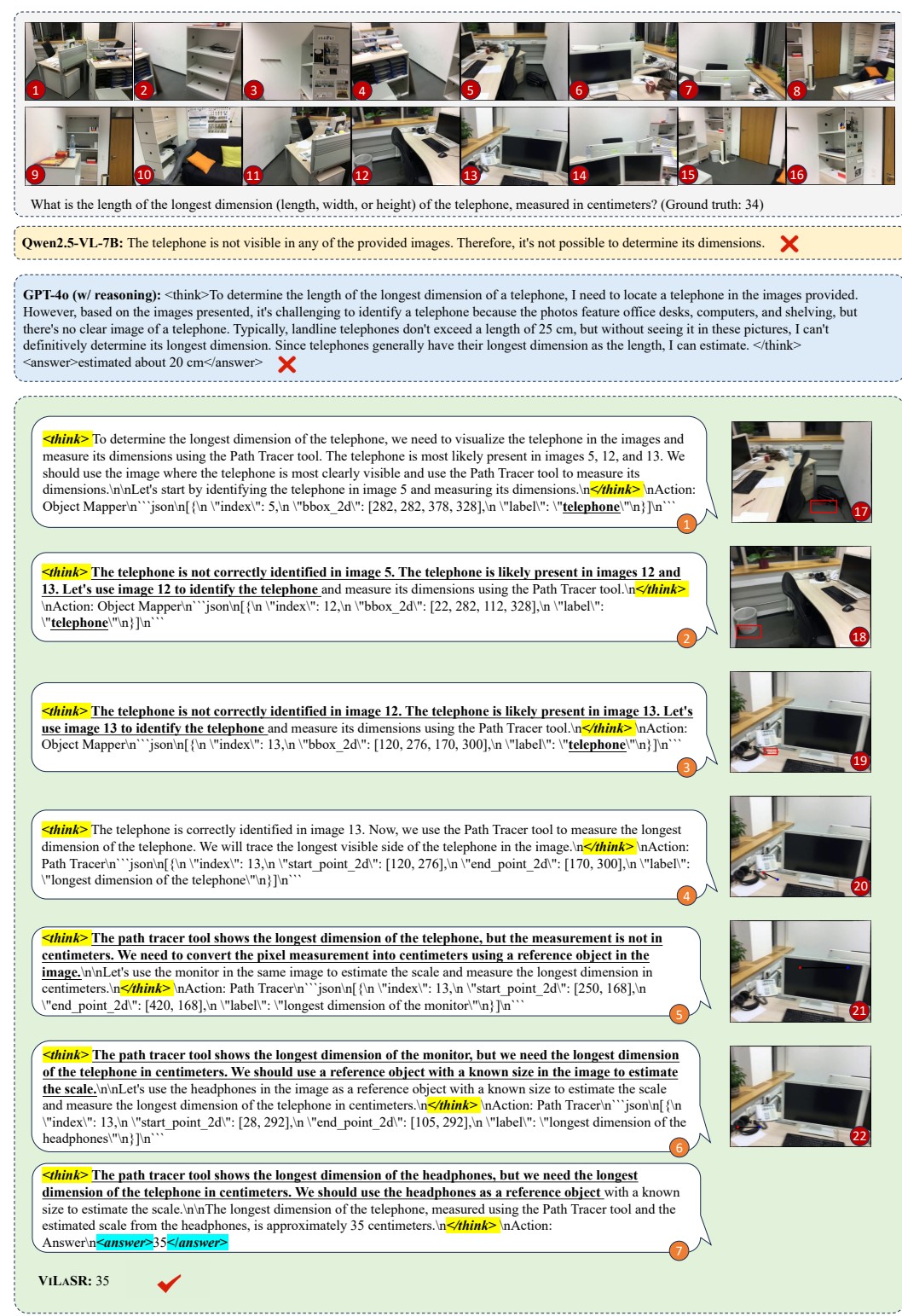

What is the length of the longest dimension (length, width, or height) of the telephone, measured in centimeters? (Ground truth: 34)

**Qwen2.5-VL-7B:** The telephone is not visible in any of the provided images. Therefore, it's not possible to determine its dimensions. ✗

**GPT-4o (w/ reasoning):** <think>To determine the length of the longest dimension of a telephone, I need to locate a telephone in the images provided. However, based on the images presented, it's challenging to identify a telephone because the photos feature office desks, computers, and shelving, but there's no clear image of a telephone. Typically, landline telephones don't exceed a length of 25 cm, but without seeing it in these pictures, I can't definitively determine its longest dimension. Since telephones generally have their longest dimension as the length, I can estimate. </think> <answer>estimated about 20 cm</answer> ✗

*<think>* To determine the longest dimension of the telephone, we need to visualize the telephone in the images and measure its dimensions using the Path Tracer tool. The telephone is most likely present in images 5, 12, and 13. We should use the image where the telephone is most clearly visible and use the Path Tracer tool to measure its dimensions.\n\nLet's start by identifying the telephone in image 5 and measuring its dimensions.\n*</think>* \nAction: Object Mapper\n```json\n[{\n \"index\": 5,\n \"bbox_2d\": [282, 282, 378, 328],\n \"label\": \"**telephone**\"\n}]\n``` ①

*<think>* **The telephone is not correctly identified in image 5. The telephone is likely present in images 12 and 13. Let's use image 12 to identify the telephone** and measure its dimensions using the Path Tracer tool.\n*</think>* \nAction: Object Mapper\n```json\n[{\n \"index\": 12,\n \"bbox_2d\": [22, 282, 112, 328],\n \"label\": \"**telephone**\"\n}]\n``` ②

*<think>* **The telephone is not correctly identified in image 12. The telephone is likely present in image 13. Let's use image 13 to identify the telephone** and measure its dimensions using the Path Tracer tool.\n*</think>* \nAction: Object Mapper\n```json\n[{\n \"index\": 13,\n \"bbox_2d\": [120, 276, 170, 300],\n \"label\": \"**telephone**\"\n}]\n``` ③

*<think>* The telephone is correctly identified in image 13. Now, we use the Path Tracer tool to measure the longest dimension of the telephone. We will trace the longest visible side of the telephone in the image.\n*</think>* \nAction: Path Tracer\n```json\n[{\n \"index\": 13,\n \"start_point_2d\": [120, 276],\n \"end_point_2d\": [170, 300],\n \"label\": \"longest dimension of the telephone\"\n}]\n``` ④

*<think>* **The path tracer tool shows the longest dimension of the telephone, but the measurement is not in centimeters. We need to convert the pixel measurement into centimeters using a reference object in the image.**\n\nLet's use the monitor in the same image to estimate the scale and measure the longest dimension in centimeters.\n*</think>* \nAction: Path Tracer\n```json\n[{\n \"index\": 13,\n \"start_point_2d\": [250, 168],\n \"end_point_2d\": [420, 168],\n \"label\": \"longest dimension of the monitor\"\n}]\n``` ⑤

*<think>* **The path tracer tool shows the longest dimension of the monitor, but we need the longest dimension of the telephone in centimeters. We should use a reference object with a known size in the image to estimate the scale.**\n\nLet's use the headphones in the image as a reference object with a known size to estimate the scale and measure the longest dimension of the telephone in centimeters.\n*</think>* \nAction: Path Tracer\n```json\n[{\n \"index\": 13,\n \"start_point_2d\": [28, 292],\n \"end_point_2d\": [105, 292],\n \"label\": \"longest dimension of the headphones\"\n}]\n``` ⑥

*<think>* **The path tracer tool shows the longest dimension of the headphones, but we need the longest dimension of the telephone in centimeters. We should use the headphones as a reference object** with a known size to estimate the scale.\n\nThe longest dimension of the telephone, measured using the Path Tracer tool and the estimated scale from the headphones, is approximately 35 centimeters.\n*</think>* \nAction: Answer\n*<answer>* 35 *</answer>* ⑦

**VILASR:** 35 ✓

Figure 10: A visualization example of spatial reasoning approaches on the VSI-Bench benchmark, including Qwen-2.5-VL-7B, GPT-4o, and VILASR. We highlight the generated words of VILASR that exhibit reflection behavior in **bold**.

Table 7: Average runtime comparison in seconds per sample (VILASR vs. backbone) across different benchmarks.

| Method | Average Runtime (s/sample) | | |
|---|---|---|---|
| | Maze | EmbSpatial | VSI-Bench |
| Qwen2.5-VL-7B (COT) | 1.0 | 0.9 | 1.9 |
| VILASR | 4.4 | 1.2 | 7.8 |

where $L_k$ is the length of the reasoning path generated at step $k$, and $M_k$ is the additional multimodal context incorporated at step $k$.

Assuming each reasoning step has approximately equal length, i.e., $L_i \approx L$, $M_i \approx M$ for all $i$, we can approximate:

$$T = \mathcal{O}\left(S^2 \cdot L \cdot (L+M) + S \cdot (L+M)(L+N)\right). \tag{10}$$

### E.2 Practical runtime

To provide empirical evidence for our complexity analysis, we evaluate the runtime performance of VILASR and compare it against the backbone model, Qwen2.5-VL-7B (CoT), across various benchmarks. Table 7 illustrates the average runtime time per sample.

## F Limitations

While our work demonstrates promising results in spatial reasoning tasks, several limitations warrant discussion:

First, our training framework primarily focuses on multiple-choice and numerical questions due to their amenability to automated evaluation, excluding more complex spatial reasoning scenarios that require free-form textual descriptions (e.g., detailed motion trajectory analysis). This limitation constrains the model's ability to handle open-ended spatial reasoning tasks.

Second, a potential risk in our reinforcement learning stage is the coupling between the reward function and the final evaluation metrics. Since our reward is directly tied to achieving the correct final answer on specific benchmarks, the model might learn to exploit dataset-specific shortcuts to maximize its reward, rather than developing a truly generalizable spatial reasoning ability. This is a common challenge in reward-based optimization that warrants further investigation into more process-oriented reward schemes.

Third, the effectiveness of our drawing operations is inherently limited by the 2D nature of the visual interface. Complex 3D spatial relationships and viewpoint changes, which are common in real-world scenarios, may not be adequately captured by our current drawing primitives. This limitation particularly affects the model's performance on tasks involving 3D object relationships or dynamic camera movements.

Finally, the computational cost of our three-stage training pipeline, especially during the reinforcement learning stage, may limit its accessibility to researchers with limited computational resources.

## G Broader impact

Our work on enhancing spatial reasoning capabilities in vision-language models has both positive and potential negative societal implications that warrant careful consideration.

On the positive side, improved spatial reasoning capabilities could significantly benefit various applications in robotics, autonomous navigation, and assistive technologies. For instance, more accurate spatial understanding could help robots better navigate complex environments and assist people with visual impairments in daily tasks. In educational settings, these models could provide

interactive tools for teaching spatial concepts and geometric reasoning. Additionally, the interpretable nature of our drawing-based reasoning approach enhances model transparency, potentially increasing trust and adoption in critical applications.

However, several potential negative impacts need to be addressed. First, the enhanced spatial reasoning capabilities could be misused for surveillance purposes, enabling more sophisticated tracking and monitoring systems that could infringe on privacy rights. Second, there might be accessibility issues as the drawing-based reasoning approach assumes users have access to and can interact with visual interfaces, potentially excluding certain user groups.

To mitigate these concerns, we recommend: (1) implementing strict usage guidelines and access controls when deploying these models in sensitive applications, (2) exploring alternative interaction modalities to make the technology more accessible. We also encourage future research to focus on developing privacy-preserving spatial reasoning techniques.

To ensure responsible deployment of our technology, we have implemented several safeguards in our release. First, our model access will be gated through an API that requires user agreement to usage guidelines, specifically prohibiting applications in surveillance or privacy-invasive systems. Second, we provide detailed documentation about the model's capabilities and limitations, along with best practices for responsible implementation in different application scenarios.

