# OpenReview forum: "Reinforcing Spatial Reasoning in Vision-Language Models with Interwoven Thinking and Visual Drawing"
_NeurIPS.cc/2025/Conference — NeurIPS 2025 poster_

### Official Review · Reviewer_ZTi9 · 2025-07-02

**Clarity:** 3
**Significance:** 3
**Originality:** 3
**Rating:** 5
**Confidence:** 3

**Summary:**

This paper introduces a new model called SPARK, leveraging ‘drawing to reason in space’, a vision centric alternative to the tex-only chain of thought that current large vision-language models (LVLMs) use. SPARK is a VLM (built on a Qwen-2.5-VL-7B backbone) that combine natural language thinking with 2 explicit visual operations: i) drawing bonding boxes and ii) drawing auxiliary lines. With the 2 additional mechanisms, SPARK can externalize spatial hypothesis directly on the image (or video frames). SPARK is trained in 3 steps: i) Cold start supervised fine-tuning (to teach basic drawing skills), ii) reflective rejection sampling and iii) a Reinforcement Learning training. SPARK is evaluated in 5 benchmarks covering static images, videos and multi-view scenes. SPARKS consistently outperforms open-source LVLMs (up to 72B) and specialized reasoning systems.

**Questions:**

1. Do you think it would be possible to extend the operation set (e.g. drawing polygone, heat maps) without retraining from scratch ? If so, please discuss and what kind of tasks you expect improvement
2. There is no systematic study on how SPARK is sensitive to video length (or image resolution) ? Does it scale well to long video sequence (or very high resolution images)
3. Have you tried different backbone (e.g. larger backbone like Qwen 72B). Do you think SPARK could easily scale up ? And also do you think it will work with vision encoder (e.g. CLIP-VIT)
4. What is the average wall-clock latency per chain-of-thought compared with classical text-only models ?
5. Does reflective rejection hurt exploration by discarding partially correct, but creative trajectories ?

**Ethical Concerns:**

["NO or VERY MINOR ethics concerns only"]

**Final Justification:**

I still think this article deserves to be accepted : strong methodology, real innovation, an potential high impact !
I keep my rating (5/6)

**Limitations:**

The main limitation lies in the fact that evolution is restricted to categorical (and numerical) QA. There is no assessment of free-form explanations or open-ended planning tasks.

**Paper Formatting Concerns:**

No concerns

**Quality:**

3

**Strengths And Weaknesses:**

Strengths: The Vision-centred reasoning paradigm is novel and well motivated: by allowing the model to draw spatial artifacts, the model aligned more closely with human spatial cognition. The 3 stages curriculum is well motivated and also empirically validated (thrgouht ablation studies). The benchamarks are broad enough to convincingly demonstrate the performance of SPARK. I have appreciated the interpretable outputs: the drawn artifacts make the model’s reasoning chain verifiable by humans, which is a useful property for safety and debugging.

Weaknesses:
1. Only boxes and straight lines are supported; many spatial tasks (3D rotation, occlusion…) might need richer geometric primitives to be solved.
2. The training and the evaluation avoid free-form answers; realisms and linguistic generalization remain untested.
3. Concerning the reward design, it seems that the same correctness/MRA formula drives both the optimisation and the test evaluation… I am concerned because there is a huge risk of overfitting…
4. There is not quantification of the computation overhead: drawing and image re-encoding each step adds inference latency and memory, would be good to at least quantify what it the additional compute/overhead.

---

> ### Author Rebuttal · Authors · 2025-07-31
>
> > [W1] Only boxes and straight lines are supported; many spatial tasks (3D rotation, occlusion…) might need richer geometric primitives to be solved. [Q1] Do you think it would be possible to extend the operation set (e.g. drawing polygone, heat maps) without retraining from scratch ? If so, please discuss and what kind of tasks you expect improvement
>
> We appreciate the reviewers' insightful questions. Boxes and straight lines are currently used as lightweight yet effective primitives to support spatial reasoning. To further enhance SPARK’s capability, we consider **two categories of extended operations**:
>
> 1.  **Operations that provide additional spatial cues**, such as magnification, heatmaps, or depth maps. These can often be incorporated **without retraining**, by directly integrating them as new tools in the drawing pipeline.
>
> 2.  **Operations that require the model to reason with new geometric representations**, such as 3D bounding boxes. These do not introduce external information but guide the model to **focus on important regions** and perform **spatial exploration and grounded reasoning**. Such capabilities typically require **continued fine-tuning** with additional demonstrations to be used effectively.
>
> We expect such extensions to be especially beneficial for **complex 3D spatial reasoning tasks**, including mental rotation, occlusion handling, and perspective inference.
>
>
> > [W2] The training and the evaluation avoid free-form answers; realisms and linguistic generalization remain untested. [W3] Concerning the reward design, it seems that the same correctness/MRA formula drives both the optimisation and the test evaluation… I am concerned because there is a huge risk of overfitting…
>
> We appreciate the comment. Most benchmark tasks use structured outputs (e.g., choices or numbers) to enable **objective and reproducible evaluation**, and to focus on **core spatial reasoning** rather than linguistic variability. This design is aligned with existing spatial benchmarks, which emphasize **decision-making over language generation**.
>
> We agree that assessing **linguistic generalization** is valuable, and our framework is compatible with extensions to free-form tasks (e.g., spatial description or explanation), which we plan to explore in future work.
>
>
> > [W4]  There is not quantification of the computation overhead: drawing and image re-encoding each step adds inference latency and memory, would be good to at least quantify what it the additional compute/overhead. [Q4] What is the average wall-clock latency per chain-of-thought compared with classical text-only models ?
>
> We thank the reviewers for raising this important point. We provide a quantification of SPARK’s computational overhead across benchmarks, including both average wall-clock runtime per sample and average number of reasoning steps:
>
> | | Maze | EmbSpatial | VSIbench |
> |---------------|------|------------|----------|
> | **Avg. Runtime of Qwen2.5-VL-7B w/ Reasoning** | 1.0  | 0.9        | 1.9     |
> | **Avg. Runtime of SPARK** | 4.4  | 1.2        | 7.8      |
>
> Compared to classical text-only CoT models, SPARK introduces additional latency primarily due to drawing and image re-encoding at each step. The runtime grows with reasoning depth and task complexity.
> Relevant results are presented in Appendix C.2 of the supplementary material. We will ensure that the appendix is clearly organized to facilitate understanding for readers.
>
> > [Q2] There is no systematic study on how SPARK is sensitive to video length (or image resolution) ? Does it scale well to long video sequence (or very high resolution images)
>
> we examined how variations in maximum image/frame resolution and the number of frames impact the model's performance. The results are summarized in the table below:
>
> | Max Resolution  | Number of Frames | VSIbench|
> |--------|----------|----------|
> |256\*28\*28| 16 | 42.9|
> |448\*28\*28|16| 43.3 |
> |256\*28\*28|32| 43.7|
>
> As shown, increasing either the number of frames or the resolution results in a slight performance improvement. This suggests that higher frame resolution and a greater number of frames contribute positively to the model's reasoning ability.
> We will include these new experimental results in our revised manuscript.
>
>
> > [Q3] Have you tried different backbone (e.g. larger backbone like Qwen 72B). Do you think SPARK could easily scale up ? And also do you think it will work with vision encoder (e.g. CLIP-VIT)
>
> Thank you for your question. Our reasoning paradigm is architecture-agnostic and can be directly scaled across various LVLMs **without fundamental modifications**. While we have not yet trained SPARK on the Qwen2.5-VL-72B backbone due to computational resource constraints, we have utilized Qwen 72B for inference and cold-start data construction. We anticipate that performance will improve steadily with scale, as observed in many spatial reasoning tasks.
>
> Our reasoning paradigm modifies only the inference procedure, without altering the model architecture itself. Therefore, it is generally applicable to **other LVLM architectures**, and is **agnostic to the specific choice of LLM or vision encoder**. As for the specific vision encoder mentioned (e.g., CLIP-ViT), we are not certain about its exact integration details within a given LVLM pipeline, but in principle, SPARK can operate with any LVLMs.
>
> > [Q5] Does reflective rejection hurt exploration by discarding partially correct, but creative trajectories ?
>
> We appreciate this insightful question about the potential impact of reflective rejection sampling on exploration and creativity. However, we argue that our approach does not hurt exploration diversity because our rejection mechanism only filters based on the presence of self-correction behaviors, not their specific forms or reasoning paths. This allows the model to discover diverse ways of revising its reasoning while maintaining accuracy.
>
> Furthermore, our quantitative analysis suggests that our approach actually enhances trajectory diversity. Specifically, we analyze the diversity of reasoning trajectories over 500 generated examples on VSIBench using the **distinct-3/4 scores**, which increased from 0.2134/0.3262 to 0.2148/0.3359 after the reflective rejection sampling stage. These results suggest that reflective rejection sampling not only **preserves exploratory behavior** but also leads to **a modest improvement in trajectory diversity**.

---

> > ### Comment · Reviewer_ZTi9 · 2025-08-05
> > **Response to rebuttal**
> >
> > Thank you for the clarifications ! I still think this article deserves to be accepted, I will therefore keep my score !

---

> > > ### Author Response · Authors · 2025-08-06
> > >
> > > We sincerely thank you for your thoughtful consideration and continued support of our work! We are truly grateful for your recognition of our contribution and for your positive evaluation. Your encouraging feedback has been greatly appreciated, and we are glad that our clarifications were helpful. Thank you again for your time and insightful review!

---

### Official Review · Reviewer_nXQB · 2025-07-02

**Clarity:** 2
**Significance:** 3
**Originality:** 3
**Rating:** 4
**Confidence:** 4

**Summary:**

This paper introduces SPARK to enhance the spatial reasoning capabilities of LVLMs. Moving beyond the direct application of text-based reasoning to visual tasks, SPARK proposes a "drawing to reason in space" paradigm. By employing tools in a manner analogous to LLM agents, SPARK generates visual and textual reasoning paths. The progressive three-stage training framework—comprising cold-start training, reflective rejection sampling, and reinforcement learning—enables substantial performance improvements across multiple spatial reasoning benchmarks.

**Questions:**

- Q1. The Reasoning Process: Are the reasoning steps generated autonomously by the model? If so, what termination conditions are in place to prevent infinite or non-productive reasoning loops?
- Q2. The Attribution of Performance Gains: Given that SPARK was fine-tuned on specialized data while the baselines were not, how can the performance gains be attributed specifically to the "drawing to reason" paradigm versus the data advantage? Can the ablation study be used to help disentangle these two factors?
- Q3. The Cost-Benefit of the Iterative Method: Considering the high computational cost of SPARK's iterative process, what are its specific advantages over simpler strategies that also use a larger compute budget (e.g., textual self-correction, sample-and-vote)? What justifies the added complexity of the visual iteration mechanism?
- Q4. Code and Data Reproducibility: The paper's checklist states that code and data are in the supplementary material, but they could not be located. Could the authors please clarify the access method to ensure reproducibility?

**Ethical Concerns:**

["NO or VERY MINOR ethics concerns only"]

**Final Justification:**

The visual drawing-based reasoning is sound and effectively verfied by experiments.

**Limitations:**

Yes

**Quality:**

3

**Strengths And Weaknesses:**

Strengths
- S1. Paradigm Innovation: The paper's "drawing to reason in space" paradigm innovates upon previous methods that apply text-centric reasoning to visual tasks by empowering the model to reason through direct drawing operations in the visual space.
- S2. High Interpretability: The method generates a "visual chain of thought," presenting the reasoning process as a transparent sequence of interleaved images and text. This significantly enhances interpretability and trustworthiness.

Weaknesses
- W1. Limited Technical Novelty of Training Framework: The proposed three-stage training framework is primarily a sequential application of existing, widely-used methods (cool-start and RL) and may lack sufficient novelty to be a core contribution.
- W2. Confounding Factors in Performance Gains: It is difficult to attribute the model's strong performance solely to the "drawing to reason" paradigm, as it benefits from specialized fine-tuning data that most baseline models did not have access to.
- W3. Lack of Comparisoin on Computational Cost: The method's high iterative cost creates an unfair comparison against the single-pass baseline models. The experiments lack a control group under an equivalent computational budget (e.g., allowing for multi-sample voting), which makes it difficult to assess the intrinsic effectiveness of the paradigm itself.

---

> ### Author Rebuttal · Authors · 2025-07-31
>
> Thank you for your valuable reviews. We provide detailed responses to address your concerns:
>
> > [W1] Limited Technical Novelty of Training Framework: The proposed three-stage training framework is primarily a sequential application of existing, widely-used methods (cool-start and RL) and may lack sufficient novelty to be a core contribution.
>
> We respectfully disagree with this assessment. Our work makes several significant contributions:
>
> -   Analysis of Current Limitations: Through careful analysis, we identify a fundamental flaw in existing multimodal reasoning approaches: while current LVLMs excel at basic visual tasks like object detection and narrative understanding, they rely solely on text-based reasoning chains, with the only difference from text-only tasks being the presence of multimodal input. This text-centric approach assumes that visual information can be perfectly translated into textual semantic space. However, such translation may inevitably lose critical spatial details - geometric relationships become ambiguous when described in text, and tracking dynamic changes of object positions becomes prohibitively complex, as evidenced by the poor performance of current systems.
> -   Novel Vision-Centric Reasoning Paradigm: To address the limitations of current LVLMs, we propose that multimodal reasoning should dynamically incorporate new visual features throughout the reasoning process, leading to our "draw to reason in space" framework. While this conceptually aligns with OpenAI's recent "thinking with images" paradigm (April 2025), our work was conducted independently and addresses more challenging scenarios that require understanding spatial relationships across multiple video frames. Moreover, unlike recent tool-based approaches that rely on black-box perception tools (e.g., grounding and OCR systems) or heavily draw upon offline-curated reasoning data, our framework enables more flexible and intrinsic visual reasoning.
> -   Technical Innovation in Training: While our framework builds on established methods like cold-start training and reinforcement learning, we introduce a novel reflective rejection sampling stage. This innovation significantly enhances the model's self-reflection capabilities (from 4.3% to 12.5% reflection rate) and improves the potential for subsequent RL optimization, as demonstrated in our ablation studies.
> -   Strong Empirical Validation: Our comprehensive experiments demonstrate substantial improvements over text-only reasoning baselines across diverse spatial tasks, with an average improvement of 11.5% compared wtih the backbone model. This empirical evidence strongly supports our core thesis that vision-centric reasoning is essential for effective spatial understanding.
>
> In summary, while individual components may build on existing techniques, our work's novelty lies in identifying a critical limitation in current approaches and proposing a principled solution supported by both theoretical analysis and strong empirical results.
>
> >[W2] Confounding Factors in Performance Gains: It is difficult to attribute the model's strong performance solely to the "drawing to reason" paradigm, as it benefits from specialized fine-tuning data that most baseline models did not have access to. [Q2] The Attribution of Performance Gains: Given that SPARK was fine-tuned on specialized data while the baselines were not, how can the performance gains be attributed specifically to the "drawing to reason" paradigm versus the data advantage? Can the ablation study be used to help disentangle these two factors?
>
> We appreciate the reviewer’s insightful question. To address this concern, we conducted a controlled ablation study in which we fine-tuned the same base model Qwen2.5-VL-7B using only question-answer pairs, without any intermediate reasoning supervision—constructing a standard VQA-style SFT pipeline. The results are shown below:
>
> | Method               | Maze | SpatialEval | EmbSpatial | VSIbench | Spar-Bench |
> |----------------------|------|-------------|------------|----------|------------|
> | Qwen2.5-VL-7B + SFT     | 73.8 | 60.4        | 61.3       | 39.1     | 33.7       |
> | SPARK                | 89.0 | 63.4        | 64.9       | 42.9     | 34.9       |
>
> These results clearly show that incorporating 'drawing to reason in space' leads to substantial improvements across all benchmarks—particularly on tasks like Maze and VSIbench, which demand explicit modeling of spatial layout and sequential planning.
>
> These results demonstrate that **drawing operations play a crucial role in spatial reasoning beyond what can be achieved through standard fine-tuning**. While both approaches were trained on the same dataset, SPARK's ability to dynamically introduce novel spatial relationships through drawing operations leads to substantial improvements. This validates our core thesis that effective spatial reasoning requires direct visual manipulation rather than purely textual reasoning.
>
> We will include the comparison in our revised revision.
>
> > [W3] Lack of Comparisoin on Computational Cost: The method's high iterative cost creates an unfair comparison against the single-pass baseline models. The experiments lack a control group under an equivalent computational budget (e.g., allowing for multi-sample voting), which makes it difficult to assess the intrinsic effectiveness of the paradigm itself. [Q3] The Cost-Benefit of the Iterative Method: Considering the high computational cost of SPARK's iterative process, what are its specific advantages over simpler strategies that also use a larger compute budget (e.g., textual self-correction, sample-and-vote)? What justifies the added complexity of the visual iteration mechanism?
>
> We incorporated the two suggested baselines: **Self-Correction** and **Multi-Sample Voting**, with implementation details as follows.
>
> In **Self-Correction** [1], the model first generates an answer with reasoning, then critiques and refines it based on its own output. This cycle repeats until no further changes are made, allowing for iterative self-improvement.
>
> In **Multi-Sample Voting** [2], we sample five reasoning trajectories and answers independently, and then adopt majority voting to determine the final answer.
>
> **Performance Comparison**
>
> | Method             | Maze | SpatialEval | EmbSpatial | VSIbench | Spar-Bench |
> |--------------------|------|-------------|------------|----------|------------|
> |Qwen2.5-VL-7B| 33.7|58.5 | 47.9 | 32.7  | 31.2 |
> | Qwen2.5-VL-7B w/ Reasoning  | 36.5  |    56.3    |  52.7   | 26.2  |  31.6  |
> | Self-correction | 36.9  |    59.8    | 55.8    | 26.6   |  31.5    |
> | Multi-Sample Voting | 32.5  |    55.3    | 57.2    | 25.8   | 30.4     |
> | SPARK  | 89.0 | 63.4        | 64.9       | 42.9     | 34.9       |
>
> As shown, both self-correction and multi-sample voting improve upon standard prompting by leveraging additional computation through error correction or output aggregation, respectively. However, these methods are fundamentally limited to textual processing and cannot incorporate visual feedback across iterations.
>
> SPARK's iterative visual reasoning framework refines spatial understanding through repeated visual grounding and feedback loops, consistently outperforming baselines across diverse spatial reasoning benchmarks.
>
> **Computational Overhead Analysis**
>
> We also report the computational overhead for SPARK and the text-based baselines in terms of average wall-clock runtime per sample:
>
> | | Maze | EmbSpatial | VSIbench |
> |---------------|------|------------|----------|
> |**Avg. Runtime of Qwen2.5-VL-7B w/ Reasoning**| 1.0|0.9|1.9|
> |**Avg. Runtime of Self-Correction**| 3.5|2.1| 5.3|
> |**Avg. Runtime of Multi-Sample Voting**|2.0|1.5|3.4|
> | **Avg. Runtime of SPARK** | 4.4  | 1.2        | 7.8      |
>
> While SPARK introduces higher computational cost due to its iterative drawing to reason, this overhead is justified by the significant performance gains. Importantly, SPARK is orthogonal to both Self-Correction and Multi-Sample Voting, and can be integrated with these approaches for further improvement.
>
> [1] Self-Refine: Iterative Refinement with Self-Feedback. NIPS 2023
>
> [2] Self-Consistency Improves Chain of Thought Reasoning in Language Models. ICLR 2023
>
> > [Q1] The Reasoning Process: Are the reasoning steps generated autonomously by the model? If so, what termination conditions are in place to prevent infinite or non-productive reasoning loops?
>
> Yes, the reasoning process in SPARK is fully autonomous. It proceeds iteratively, with the model generating intermediate steps until it produces a final answer enclosed within the special tokens `<answer>` and `</answer>`, which serves as an explicit termination signal, as exemplified in Figure 1 of the current submission. Additionally, we also impose predefined upper bounds on both the **number of reasoning iterations** and the **number of accumulated sketches/images**. If either threshold is reached before an answer is generated, the process is forcefully terminated.
>
> The details of this temination condition can be found in specifically lines 158–159 and lines 214–217. We will revise and clarify the relevant descriptions in the final version.
>
> > [Q4] Code and Data Reproducibility: The paper's checklist states that code and data are in the supplementary material, but they could not be located. Could the authors please clarify the access method to ensure reproducibility?
>
> We sincerely apologize for the confusion. We failed to prepare the anonymized code and data package in time for inclusion in the supplementary material.
>
> As the rebuttal phase prohibits the inclusion of external links, we are currently unable to provide access to the code and data. However, we are fully committed to releasing the complete code and data, to ensure full reproducibility and facilitate future research.

---

> > ### Comment · Reviewer_nXQB · 2025-08-06
> >
> > After reading the authors' response and comments from other reviewers, my concerns are addressed well. Although introducing complexity, the visual drawing-based reasoning is sound and effectively verfied by experiments. I will raise my score.

---

> > > ### Author Response · Authors · 2025-08-07
> > >
> > > We are delighted that our responses have fully addressed your concerns! We sincerely appreciate the time and thoughtful consideration you have devoted to reviewing our submission. We are committed to incorporating the suggested improvements into the revised manuscript.

---

> ### Comment · Area_Chair_ne9C · 2025-08-05
>
> @Reviewer nXQB: could you please let us know whether the authors have addressed your concerns? I am especially keen to hear your feedback regarding novelty and the authors' ablation study.

---

### Official Review · Reviewer_H5eA · 2025-07-03

**Clarity:** 2
**Significance:** 1
**Originality:** 2
**Rating:** 3
**Confidence:** 5

**Summary:**

This paper introduces the SPARK framework, a vision language model that can draw to reason in space. The proposed SPARK has basic drawing operations like annotating bounding boxes and drawing auxiliary lines. Trained with a three-stage paradigm: cold start, reflective rejection sampling, and GRPO-like RL, the model can acquire capabilties to perform spatial reasoning tasks.

Evaluated on multiple spatial reasoning benchmarks spanning maze navigation, static spatial reasoning, video-based temporal tracking, and multi-view scene understanding, the model achieved good performance compared to base VL models.

**Questions:**

re weaknesses: What if you only perform query-answer (as in standard VQA) fine-tuning? Does the drawing really improve the performance? How does this compare to visual programming-like approaches for solving spatial relationship tasks [A, B]?

**Ethical Concerns:**

["NO or VERY MINOR ethics concerns only"]

**Final Justification:**

After reading the authors' response, I have decided to increase my score by one.

**Limitations:**

yes

**Paper Formatting Concerns:**

All good.

**Quality:**

2

**Strengths And Weaknesses:**

#### Strengths:

- The paper is well-written, and the idea of drawing facilitates visual reasoning is intriguing. This paper successfully makes some good attempts at letting large language models draw auxiliary lines in visual space to help their reasoning.

- The three-stage paradigm is up-to-date with contemporary advances in RL for LLM reasoning. It's good to see that it can be scaled to more operators.

#### Weaknesses:

- The drawing concept is a bit bluffing. Here, the drawing defined in this paper is merely bounding boxes and simple lines -- which have limited usage cases beyond simple spatial reasoning.

- Lacking baseline comparisons for spatial reasoning involving tool use: how does this compare to visual programming-like approaches for solving spatial relationship tasks [A, B]?

- Ablations: It seems that stage 1 is the most important stage. I suspect that the first step is more like some fine-tuning on the spatial reasoning tasks, making it unfair to compare to other base VL models. What if you only perform query-answer (as in standard VQA) fine-tuning? Does the drawing really improve the performance?

ref:

[A]. Gupta, T., & Kembhavi, A. (2023). Visual programming: Compositional visual reasoning without training. In Proceedings of the IEEE/CVF Conference on Computer Vision and Pattern Recognition (pp. 14953-14962).

[B]. Surís, D., Menon, S., & Vondrick, C. (2023). Vipergpt: Visual inference via python execution for reasoning. In Proceedings of the IEEE/CVF International Conference on Computer Vision (pp. 11888-11898).

---

> ### Author Rebuttal · Authors · 2025-07-31
>
> Thank you for your valuable reviews. We first address Weakness 2 and Weakness 3, and then respond to Weakness 1. Below are our detailed responses:
>
> > [W2] Lacking baseline comparisons for spatial reasoning involving tool use: how does this compare to visual programming-like approaches for solving spatial relationship tasks [A, B]?
>
> We appreciate this suggestion. We have incorporated the two recommended baselines — **VisProg** and **ViperGPT** — into our experiments. The results are summarized below:
>
> | Method     | Maze | SpatialEval | EmbSpatial | VSIbench | Spar-Bench |
> |------------|------|-------------|------------|----------|------------|
> | VisProg    | 22.6   |     18.0    |  13.5    |    -     |    -       |
> | ViperGPT   |  27.3   |    34.1        |    39.0       |    10.6    |    23.7      |
> | SPARK      | 89.0 | 63.4        | 64.9       | 42.9     | 34.9       |
>
> Note: VisProg does not support multi-image inputs, making it inapplicable to video-based (VSIbench) and multi-view (Spar-Bench) tasks.
>
> As shown, SPARK achieves significantly stronger and more stable results across benchmarks compared with VisProg and ViperGPT. Furthermore, during our reimplementation of **VisProg** and **ViperGPT**, we observed their following limitations:
>
> -   **Dependency on Black-box Perception Tools:** VisProg and ViperGPT rely heavily on black-box perception modules (e.g., object detection and localization systems), whose capabilities are fixed and cannot be optimized through end-to-end training. These predefined tools not only limit the system's ability to handle complex spatial relationships, but also lead to fragmented reasoning processes through disconnected tool invocations.
> -   **Limited Robustness and Generalization**: Both methods exhibit significant brittleness in handling complex scenarios:
>
> 	- **Heavy reliance on carefully crafted prompts**, leading to format errors and unexecutable programs when facing unseen tasks, especially in tasks like **MAZE**, which demand multi-step planning and reasoning. Many samples failed due to parsing or planning issues.
> 	-   **The tendency to fall back to simple VQA tool calls** even for complex questions and **confusion with basic spatial relationships** (e.g., above-behind vs. below-in front), aligning with the observation of recent research [1].
>
>
> In contrast, SPARK provides several key advantages:
>
> 1.  **Direct Visual Manipulation**: Through elementary drawing operations, SPARK avoids the limitations of black-box perception tools, enabling more trainable and interpretable spatial reasoning.
> 2.  **Coherent Reasoning Process**: Our approach allows for coherent reasoning processes that can be iteratively refined through self-reflection and effectively optimized through RL.
> 3.  **General and Extensible Framework**: While our current implementation focuses on basic drawing operations, the framework is readily extensible to incorporate additional visual operations for different types of spatial reasoning tasks. This flexibility is evidenced by our strong performance across diverse tasks, from single-image spatial relationship understanding to complex video-based reasoning.
>
> We will incorporate these comparisons and a more detailed discussion into the revised manuscript.
>
> [1] Visual Agentic AI for Spatial Reasoning with a Dynamic API. CVPR 2025
>
> > [W3] Ablations: It seems that stage 1 is the most important stage. I suspect that the first step is more like some fine-tuning on the spatial reasoning tasks, making it unfair to compare to other base VL models. What if you only perform query-answer (as in standard VQA) fine-tuning? Does the drawing really improve the performance?
>
> We appreciate the reviewer’s insightful question. To address this concern, we conducted a controlled ablation study in which we fine-tuned the same base model using only question-answer pairs, without any intermediate reasoning supervision—constructing a standard VQA-style fine-tuning pipeline. The results are shown below:
>
> | Method               | Maze | SpatialEval | EmbSpatial | VSIbench | Spar-Bench |
> |----------------------|------|-------------|------------|----------|------------|
> | Qwen2.5-VL-7B+ SFT     | 73.8 | 60.4        | 61.3       | 39.1     | 33.7       |
> | SPARK                | 89.0 | 63.4        | 64.9       | 42.9     | 34.9       |
>
> These results clearly show that incorporating the "drawing to reason in space" paradigm leads to substantial improvements across all benchmarks, particularly on tasks like Maze and VSIbench, which demand explicit modeling of spatial layout and sequential planning.
>
> These results demonstrate that **drawing operations play a crucial role in spatial reasoning beyond what can be achieved through standard fine-tuning**. While both approaches were trained on the same dataset, SPARK's ability to dynamically introduce novel spatial relationships through drawing operations leads to substantial improvements. This validates our core thesis that effective spatial reasoning requires direct visual manipulation rather than purely textual reasoning.
> We will include the comparison in our revised revision.
>
> > [W1] The drawing concept is a bit bluffing. Here, the drawing defined in this paper is merely bounding boxes and simple lines -- which have limited usage cases beyond simple spatial reasoning.
>
> We appreciate the reviewer's thoughtful comment. We would like to clarify several important points:
>
> -   **Operation Choice:** The basic operations (frame selection, bounding boxes, and auxiliary lines) are an intentional design choice that prioritizes interpretability and controllability while maintaining sufficient expressiveness for spatial reasoning. The visual drawing elements are extensively present in pre-training data of LVLMs, making them natural building blocks for enhancing spatial understanding. The simplicity and familiarity of these operations enable the model to focus on the core task of spatial reasoning without being distracted by complex visual manipulations.
> -   **Demonstrated Effectiveness:** Our experimental results show that these elementary operations, when strategically combined, serve as a powerful mechanism for complex spatial reasoning. SPARK achieves significant improvements across diverse challenging tasks, from maze navigation to video-based spatial relationship tracking, demonstrating that simple operations can effectively capture sophisticated spatial relationships.
> -   **Advantages Over Existing Approaches:** Current approaches to multimodal reasoning broadly fall into two categories, each with significant limitations:
>
> 	-   **Text-centric reasoning and reasoning-free approaches:** While these approaches excel at basic visual tasks, they merely add multimodal input to traditional language model reasoning. This approach assumes visual information can be perfectly translated into textual space. However, **converting visual information to text inevitably loses critical spatial details and makes it prohibitively difficult to track dynamic spatial relationships**.
> 	-   **Tool-based reasoning approaches:** These works like VisProg and ViperGPT attempt to enable vision-centric reasoning through external tools. However, they are limited by: (1) Dependency on black-box perception tools, resulting in **fixed perception capabilities and fragmented reasoning processes**; (2) **Limited ability for deep reasoning with reflection on tool outputs** due to their reliance on prompting engineering. In contrast, SPARK's drawing-based reasoning provides a more flexible and intrinsic approach to spatial understanding. leading to significant improvements over VisProg, ViperGPT, and standard VQA training as shown in the results above.
> -   **Future Extensibility:** While our current implementation focuses on two fundamental operations, this work represents an innovative first step towards vision-centric reasoning. The framework is readily extensible to incorporate additional drawing operations based on specific task requirements, while maintaining the benefits of interpretability and direct visual manipulation.
>
> Our results demonstrate that these basic operations, when properly integrated into a learning framework, can achieve sophisticated spatial reasoning capabilities while maintaining simplicity and interpretability.

---

> > ### Comment · Reviewer_H5eA · 2025-08-04
> >
> > Thanks for the authors response, and I have decided to increase my rating by one.

---

> > > ### Author Response · Authors · 2025-08-05
> > >
> > > Thank you for taking the time to review our rebuttal and for raising your score accordingly. We truly appreciate the effort and thoughtfulness you have dedicated to evaluating our submission.  We are happy to provide any further clarifications if needed.

---

> > > ### Author Response · Authors · 2025-08-07
> > >
> > > Dear Reviewer H5eA,
> > >
> > > Thank you once again for your time and effort in reviewing our work. We truly value your thoughtful and constructive feedback.
> > >
> > > With the extension of the discussion period, we would like to kindly follow up on our responses. We are encouraged that other reviewers have indicated their concerns have been satisfactorily addressed, and we hope our detailed responses have likewise addressed your concerns. If any issues remain unclear or require further clarification, we would be happy to provide additional explanations.
> > > If you are satisfied with our responses, we would be deeply grateful if you could consider a positive reassessment of our work.
> > >
> > > Thank you again for your valuable insights and kind attention. We will fully incorporate your suggested comparisons and discussions into the final revised version.
> > >
> > > Best regards,
> > >
> > > Authors of Paper 20260

---

### Official Review · Reviewer_Ve3P · 2025-07-03

**Clarity:** 2
**Significance:** 3
**Originality:** 3
**Rating:** 5
**Confidence:** 4

**Summary:**

The paper introduces SPARK, a LVLM that doing multimodal reasoning via interleaved text reasoning and visual drawing. There are two drawing operations, drawing bounding boxes and auxiliary lines, on images and videos. The training recipe includes three stages: (1) cold-start SFT (2) reflective rejection sampling (3) RL. The model, which is fine-tuned over Qwen-2.5 VL, outperform the original model by an average of 11.5%. Ablation shows the effectiveness of each approach.

**Questions:**

1. The model is getting big improvements on MAZE, but relatively small gains on some other tasks. Can you give more insights on that? What new tools/training strategies do we need to make the approach more effective on these other tasks?

2. As I mentioned in the weaknesses, it would be good to include ablation on each training stage

**Ethical Concerns:**

["NO or VERY MINOR ethics concerns only"]

**Limitations:**

yes

**Paper Formatting Concerns:**

No concern.

**Quality:**

3

**Strengths And Weaknesses:**

Strengths:
1. The motivation is clear, and the method is reasonable.
2. The approach effectively gives LVLM the ability to reason with interleaved reasoning trajectories.
3. Extensive experiments on diverse image and video tasks.

Weaknesses:
1. More analysis on the inference-time scaling would be helpful. The author only shows the difference of pass@1 and pass@8. However, a more interesting dimension is the number of tokens the model uses to answer the question. It would be great to include, for example, the number of tokens and the number of tool calls during the process of reinforcement learning.
2. More analysis on the effectiveness of each training stage would be useful. What if we remove one of the stages? How would that affect the final performance?

---

> ### Author Rebuttal · Authors · 2025-07-31
>
> Thank you for your valuable reviews. We provide detailed responses to your questions:
>
> >[W1] More analysis on the inference-time scaling would be helpful. The author only shows the difference of pass@1 and pass@8. However, a more interesting dimension is the number of tokens the model uses to answer the question. It would be great to include, for example, the number of tokens and the number of tool calls during the process of reinforcement learning.
>
> We appreciate the suggestion to analyze inference-time behavior beyond just pass@k. In response, we provide detailed statistics on the model’s response length and the number of tool calls throughout reinforcement learning (RL) training, as summarized below.
>
> Here, _response length_ refers to the total number of tokens generated **after receiving the input query**, including all model-generated tokens, image tokens used during reasoning, and template tokens.
>
> | **Step** | **Response Length** | **# Tool Calls** |
> |------|------------------|---------------|
> | 0    | 2449.05          | 5.127         |
> | 50   | 2332.45          | 5.114         |
> | 100  | 2292.01          | 5.138         |
> | 150  | 2386.39          | 4.505         |
> | 200  | 2103.77          | 4.916         |
> | 250  | 2083.14          | 4.858         |
> | 300  | 1966.65          | 4.785         |
>
> As observed, the average response length gradually decreases over the course of RL training, while the number of tool calls first drops significantly and then stabilizes. This suggests that the RL phase helps the model learn to reason more efficiently and use drawing operations more selectively. We attribute this improvement to the fact that the SFT (supervised fine-tuning) phase tends to produce overly verbose or redundant reasoning paths, which RL helps refine.
>
> Furthermore, we analyze different ablations of SPARK on the VSIbench dataset, comparing the average response length and number of tool calls:
>
> | Model                  | Response Length | # Tool Calls |
> |------------------------|------------------|---------------|
> | SPARK                  | 2233.98          | 4.503         |
> | SPARK w/o Stage 3      | 2948.82          | 5.213         |
> | SPARK w/o Stage 2      | 2125.53          | 3.829         |
> | SPARK w/o Stage 2 & 3  | 2834.07          | 5.266         |
>
> Compared to the variants without Stage 3 (RL), SPARK significantly reduces both response length and the number of tool calls, indicating that RL fine-tuning improves the efficiency of reasoning and drawing.
>
> Notably, SPARK w/o Stage 2 (i.e., without the reflective rejection sampling stage) shows a lower number of tool calls than the full model. Our manual inspection found that this variant tends to shortcut reasoning by making quick spatial judgments using auxiliary lines without sufficient self-verification. This highlights the importance of the reflective phase in encouraging deliberate and robust reasoning behaviors. We will add the analysis above in our revision.
>
> >[W2] More analysis on the effectiveness of each training stage would be useful. What if we remove one of the stages? How would that affect the final performance? [Q2] As I mentioned in the weaknesses, it would be good to include ablation on each training stage
>
> Thank you for pointing this out. We have included a comprehensive ablation study to evaluate the contribution of each training stage in the submitted version. The results are presented in Table 1 and discussed in Section 4.3 of the current submission, covering the following configurations:
>
> -   w/o Stage 3 (without RL stage)
> -   w/o Stage 2 (without reflective rejection sampling stage)
> -   w/o Stage 2 & 3 (without reflective rejection sampling stage and rl stage)
> -   w/o Stage 1, 2 & 3 (without training)
>
> Our key findings are summarized below:
>
> 1. **Stage 1 – Cold-Start Training:**
> Cold-start training establishes a crucial foundation for spatial reasoning, with significant performance gaps observed when omitted, especially on tasks requiring sophisticated reasoning beyond basic spatial relations.
>
> 2. **Stage 2 – Reflective Rejection Sampling:**
> Reflection sampling is key to the model's self-correction ability, enhancing its overall reasoning capacity by refining the inference process through reflective rejection mechanisms.
>
> 3. **Stage 3 – Reinforcement Learning with Dense Rewards:**
> Reinforcement learning with dense rewards enhances performance and optimizes the decision-making process. Additionally, the use of dense rewards allows for more fine-grained feedback in numerical prediction problems, offering greater improvement on numerical prediction tasks.
>
> These results validate the necessity of each training stage and highlight their complementary roles in achieving strong overall performance.
>
>
> > [Q1] The model is getting big improvements on MAZE, but relatively small gains on some other tasks. Can you give more insights on that? What new tools/training strategies do we need to make the approach more effective on these other tasks?
>
> This is an important observation. We attribute the performance gap primarily to differences in task diversity and complexity. The MAZE task is relatively constrained, taking place in a synthetic environment with limited spatial operations (e.g., directional movement) and a relatively small maze size. In contrast, other tasks demand more sophisticated real-world spatial understanding, involving reasoning over varied spatial relations and transformations.
>
> For instance, VSIbench includes tasks such as object counting, absolute and relative distance estimation, object and room size comparisons, route planning, and appearance order reasoning — all of which introduce significantly higher complexity.
>
> To enhance performance in these more challenging tasks, we believe several directions are promising:
>
> -   **Spatial pretraining**, to strengthen the model’s foundational understanding of spatial structures and relationships.
> -   **2D/3D reconstruction training**, to enabe the model to develop a genuine understanding of spatial structures by learning to describe and reconstruct 2D or 3D scenes during training.
> -   **Multi-view selection and joint encoding**, to prioritize informative viewpoints and effectively capture spatial relations across diverse perspectives.

---

> ### Comment · Reviewer_Ve3P · 2025-08-04
>
> Thanks authors for addressing my comments. I will keep my scores

---

> > ### Author Response · Authors · 2025-08-05
> >
> > Thank you for your positive feedback—we are pleased to know that our responses have addressed your comments. We sincerely appreciate the time and effort you have dedicated to reviewing our submission.

---

### Decision · Program_Chairs · 2025-09-17

**Decision:**

Accept (poster)

**Comment:**

The paper proposes SPARK, a vision-language model that interleaves natural-language reasoning with simple visual drawing operations (boxes, lines) to perform spatial reasoning across images and videos. A three-stage training recipe (cold-start SFT, reflective rejection sampling, RL) yields consistent gains on five spatial reasoning benchmarks and produces an interpretable “visual chain of thought.”

Reviewers found the motivation clear, the paradigm shift to vision-centric reasoning interesting, and the empirical coverage broad; they also appreciated the interpretability benefits. Main weaknesses raised early on were limited novelty in the training stack, unclear attribution of gains vs. specialized training data, missing comparisons to visual-programming baselines, compute/latency overhead, restricted drawing primitives, and lack of free-form evaluation.

In rebuttal, the authors added several baselines/controls, and analyses of inference-time scaling, termination conditions, and runtime overhead; they also reported limited sensitivity studies and clarified extensibility of the operator set and plans to release code/data upon acceptance. These additions addressed most factual concerns; some open issues remain around compute cost, the narrow operator set, and the absence of free-form outputs.

After discussion, two reviewers maintained clear “accept” positions, one moved from borderline-accept to stronger support, and one reviewer raised their score but remains below the accept threshold, primarily on novelty/scope grounds. Importantly, no fundamental methodological flaws or correctness issues were identified, and the missing baselines/controls were added, which favored the method.

Given all this, the AC recommends the paper be accepted. The authors should, in the camera-ready, (i) release code/models/data, (ii) quantify compute/latency more prominently, (iii) discuss reward-evaluation coupling risks, and (iv) outline/validate richer drawing primitives and, when feasible, a free-form evaluation.